# Structural insights in cell-type specific evolution of intra-host diversity by SARS-CoV-2

Kapil Gupta [1,2,11✉], Christine Toelzer [1,2,11], Maia Kavanagh Williamson[3,11], Deborah K. Shoemark [1,2,11], A. Sofia F. Oliveira [1,4], David A. Matthews [3], Abdulaziz Almuqrin[3], Oskar Staufer [5,6,7,8], Sathish K. N. Yadav[1,2], Ufuk Borucu[1,2], Frederic Garzoni[9], Daniel Fitzgerald[10], Joachim Spatz [5,6,7,8], Adrian J. Mulholland [4], Andrew D. Davidson[3], Christiane Schaffitzel [1,2,10✉] & Imre Berger [1,2,4,8,10✉]

As the global burden of SARS-CoV-2 infections escalates, so does the evolution of viral variants with increased transmissibility and pathology. In addition to this entrenched diversity, RNA viruses can also display genetic diversity within single infected hosts with co-existing viral variants evolving differently in distinct cell types. The BriSΔ variant, originally identified as a viral subpopulation from SARS-CoV-2 isolate hCoV-19/England/02/2020, comprises in the spike an eight amino-acid deletion encompassing a furin recognition motif and S1/S2 cleavage site. We elucidate the structure, function and molecular dynamics of this spike providing mechanistic insight into how the deletion correlates to viral cell tropism, ACE2 receptor binding and infectivity of this SARS-CoV-2 variant. Our results reveal long-range allosteric communication between functional domains that differ in the wild-type and the deletion variant and support a view of SARS-CoV-2 probing multiple evolutionary trajectories in distinct cell types within the same infected host.

[1] School of Biochemistry, University of Bristol, 1 Tankard's Close, Bristol BS8 1TD, UK. [2] Bristol Synthetic Biology Centre BrisSynBio, 24 Tyndall Ave, Bristol BS8 1TQ, UK. [3] School of Cellular and Molecular Medicine, University of Bristol, University Walk, Bristol BS8 1TD, UK. [4] School of Chemistry, University of Bristol, Cantock's Close, Bristol BS8 1TS, UK. [5] Department for Cellular Biophysics, Max Planck Institute for Medical Research, Jahnstraße 29, 69120 Heidelberg, Germany. [6] Institute for Physical Chemistry, Department for Biophysical Chemistry, University of Heidelberg, Im Neuenheimer Feld 253, 69120 Heidelberg, Germany. [7] Max Planck School Matter to Life, Jahnstraße 29, D-69120 Heidelberg, Germany. [8] Max Planck Bristol Centre for Minimal Biology, Cantock's Close, Bristol BS8 1TS, UK. [9] Imophoron Ltd, St. Philips Central, Albert Rd, St. Philips, Bristol BS2 0XJ, UK. [10] Halo Therapeutics Ltd, St. Philips Central, Albert Rd, St. Philips, Bristol BS2 0XJ, UK. [11]These authors contributed equally: Kapil Gupta, Christine Toelzer, Maia Kavanagh Williamson, Deborah K. Shoemark. ✉email: kapil.gupta@bristol.ac.uk; cb14941@bristol.ac.uk; imre.berger@bristol.ac.uk

SARS-CoV-2 spike (S) glycoprotein prominently differs from other betacoronavirus S proteins in the insertion of a furin cleavage site in the S1/S2 junction site[1]. The S trimer glycoprotein is responsible for binding to the ACE2 receptor and for viral cell entry after cleavage at the S1/S2 junction and S2′ sites[2]. Critical to this process is proteolytic processing of S by host cell proteases[3]. After intracellular cleavage at the S1/S2 junction by a furin-like protease to produce the S1 and S2 subunits, S gets destabilized and can be further primed by cleavage at the S2′ site by host serine proteases on the plasma membrane such as TMPRSS2[4,5] or the endosomal cysteine proteases cathepsin B/L[6]. S1 comprises the N-terminal domain (NTD), the receptor-binding domain (RBD), and the SD1 and SD2 domains[7,8]. S2 contains the S2′ cleavage site, the fusion peptide, a fusion peptide proximal region (FPPR), a HR1 heptad repeat, a central helix and a connector domain followed by a HR2 heptad repeat, the transmembrane domain and the C-terminal cytoplasmic domain[7,8]. Receptor binding destabilizes S, allowing S2′ cleavage, leading to shedding of S1 while S2 reorganizes to mediate fusion of viral and cellular membranes, enabling entry of SARS-CoV-2 into the host cells[9]. The furin cleavage site is a four amino acid motif located on a solvent-exposed flexible loop of S[7]. Furin-cleaved S was shown to open more efficiently suggesting an increased binding to human ACE2 than uncleaved S[10]. The furin cleavage site thus contributes substantially to the high infectivity of SARS-CoV-2, adding to the lethality of the virus.

After the growth of a low passage isolate of SARS-CoV-2 from February 2020 in the African green monkey kidney cell line Vero E6, a cell line routinely used to propagate viruses from clinical isolates, we discovered a virus subpopulation with an S variant (termed here BriSΔ) exhibiting an in-frame 8 amino acid deletion encompassing the furin recognition motif and S1/S2 cleavage site (amino acids 679-687 NSPRRARSV, replaced by I)[11]. Subsequently, further deletion variants abrogating S1/S2 cleavage were identified after viral passaging in cell culture[12–14] and at low frequency in clinical samples, attenuating infection in animal models[15–18]. Moreover, deleting only PRRA in S by reverse genetics resulted in a recombinant ΔPRRA SARS-CoV-2 which exhibited increased infectivity and viral titer in Vero E6 cells, but a 10-fold reduced viral titer in Calu-3 2B4 lung epithelial carcinoma cells compared to the wild-type (WT) virus[19], indicating the acquisition of a furin cleavage site increased SARS-CoV-2 fitness for replication in respiratory cells.

Here, we dissect the structure, dynamics and mechanism of the BriSΔ deletion variant S we identified, to gain insight into how diversification of the virus by elimination of a loop-region comprising the furin recognition motif and S1/S2 cleavage site impacts viral cell tropism, infectivity, spike protein stability and receptor binding, revealing molecular communication between functional regions within the spike glycoprotein allowing SARS-CoV-2 to evolve intra-host diversity in distinct cell types.

## Results

**BriSΔ variant and wild-type SARS-CoV-2 clonal isolation.** Direct RNA sequence analysis of a virus stock of SARS-CoV-2 isolate hCoV-19/England/02/2020, produced by a single passage in Vero E6 cells, revealed the presence of the WT SARS-CoV-2 and the BriSΔ variant (Fig. 1a). To obtain homogenous virus populations, the mixed virus stock was subjected to two rounds of limiting dilution in Vero E6 and human Caco-2 cells (Supplementary Fig. 1). Nanopore direct RNA sequencing confirmed that the limiting dilution yielded WT SARS-CoV-2 from Caco-2 cells. In contrast, BriSΔ was selected for in Vero E6 cells (Fig. 1a) as expected[11]. The differences in the infectivity of the WT and BriSΔ viruses were then compared on Vero E6, Vero E6/TMPRSS2,

Caco-2, Caco-2-ACE2 and Calu-3 cells using a range of virus dilutions for infection (Fig. 1b–f). The starting virus volumes for the infections were based on equal viral genome copy numbers as determined by qRT-PCR (equating to a starting multiplicity of infection (MOI) of 10 for the WT virus, based on the Vero E6 cell titer) rather than MOI values. Although viral genome copy numbers do not necessarily reflect virus infectivity, viral growth assays on Vero E6 and Caco-2-ACE2 cells using MOIs determined on either Vero E6 or Caco-2-ACE2 cells showed that the infectivity of the two viruses differed, depending on the cell type used to determine the MOI (Supplementary Fig. 2). The percentage of virus-infected cells for the five different cell lines was analyzed 18 hours after virus infection, before multiple rounds of virus replication. In Vero E6 cells, half-maximal infection was achieved with an ~6-fold higher dilution of BriSΔ as compared to WT virus (Fig. 1b). Overexpression of TMPRSS2 protease in Vero E6 cells[5] resulted in a substantially higher infection efficiency for both viruses; close to 100% of cells were infected with an up to 16-fold dilution of WT virus and an up to 64-fold dilution of BriSΔ virus (Fig. 1c). Thus, the lack of the TMPRSS2 protease contributes to, but is not the only reason why the BriSΔ variant infects Vero E6 cells better than WT virus. Differences in the route of cell entry either via fusion at the plasma membrane or receptor-mediated endocytosis[20–22] may account for this result. Interestingly our results using Vero E6/TMPRSS2 cells are similar to those of Zhu et al[18] comparing the replication of WT SARS-CoV-2 and a virus (Sdel) containing a 7 amino acid deletion encompassing the furin cleavage site but differ from those based on a competition assay between the WT and ΔPRRA viruses, which infected Vero E6/TMPRSS2 cells equally well[19]. Even though we selected WT SARS-CoV-2 from Caco-2 cells by serial dilution, BriSΔ infected Caco-2 cells better than the WT at high virus titers; 25% versus 10% infected cells were observed with the starting dilutions of BriSΔ and WT, respectively (Fig. 1d). Overexpression of the ACE2 receptor in Caco-2 cells led to 70% infection of cells up to 32-fold dilution of WT, whereas only about 35% of cells could be infected by the BriSΔ variant at the same dilution (Fig. 1e). Thus, both the WT and BriSΔ viruses infect Caco-2 cells better when ACE-2 expression was increased, but the improvement for WT virus was substantially higher. Calu-3 lung cells were infected about 2-fold better by WT virus for all dilutions except the starting dilutions (Fig. 1f), corroborating the contribution of the furin site in SARS-CoV-2 S to improved infection of lung cells[19]. Differences in the maximal level of infection of the different cell lines at 18 hours after virus infection were observed, most likely due to either difference in the expression levels of ACE2 and cellular proteases required for virus entry or intrinsic cellular factors restricting initial viral replication[22].

Next, we tested neutralization of the WT and BriSΔ viruses. No difference was found in neutralization of the two viruses when Vero E6/TMPRSS2 and Vero E6 cells were infected with equal amounts of infectious virus based on cell infectivity, in the presence of a commercial antibody binding the RBD (Fig. 1g) or human serum from a convalescent COVID-19 patient (Supplementary Fig. 3), respectively, indicating that both virus species were neutralized with equal potency by the antibodies.

**Cryo-EM structure of BriSΔ glycoprotein.** To understand the structural impact of the deletion of the furin cleavage site on SARS-CoV-2 S architecture, we produced the BriSΔ spike by MultiBac/insect cell expression[23]. We purified the glycoprotein by affinity purification and size exclusion chromatography (Supplementary Fig. 4a, b) We used the peak fraction from SEC for negative stain EM quality control (Supplementary Fig. 4c) and

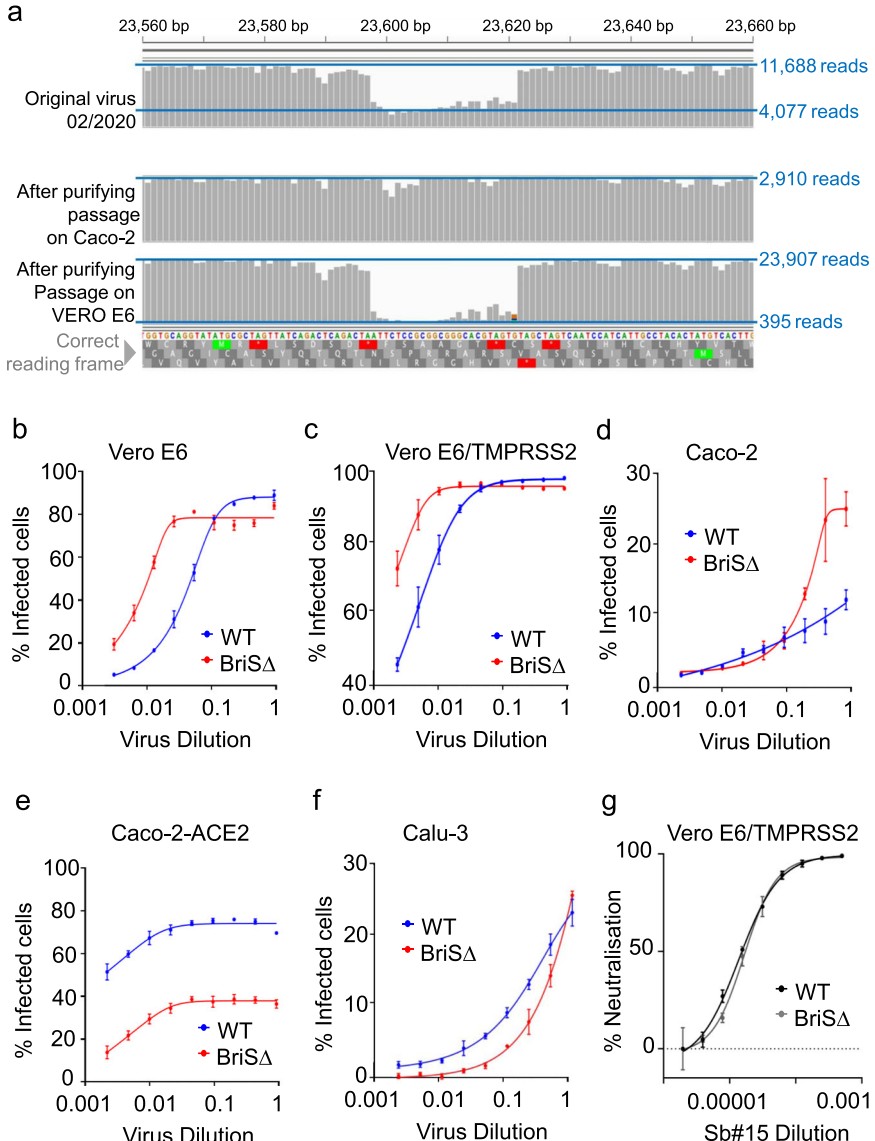

**Fig. 1 hCoV-19/England/02/2020 derived SARS-CoV-2 BriSΔ variant. a** Depth of read across S glycoprotein gene at furin cleavage site is shown for three different stocks of SARS-CoV-2 isolate hCoV-19/England/02/2020. Horizontal blue lines indicate read depth. The original stock of virus (top) evidences sharp decline in read depth corresponding to in-frame deletion of the furin cleavage site and indicative of a mixed population of viruses. The middle panel shows the sequencing depth at same region for a virus stock that has been isolated by growth on human Caco-2 cells and purified by limiting dilution. The bottom panel shows the equivalent sequencing data for a stock of the SARS-CoV-2 BriSΔ variant grown on Vero E6 cells and purified by limiting dilution. **b–f** SARS-CoV-2 infection assays: Approximately equal amounts of the WT virus and BriSΔ virus based on genome amounts (estimated by qRT-PCR) were diluted (2-fold dilution series starting with neat virus) and used to infect Vero E6, Vero E6/TMPRSS2, Caco-2, Caco-2-ACE2, and Calu-3 cells. At 18 h after infection, cells were fixed, stained with an anti-N antibody and the % of cells infected was determined by immunofluorescence microscopy. Data (**b–f**) are presented as mean values ±SD. $n = 3$ biological replicates. **g** WT virus and BriSΔ virus were used to infect Vero E6/TMPRSS2 cells in the presence of a range of dilutions of a commercial antibody against SARS-CoV-2 RBD. Cells were infected with equal amounts of infectious virus (based on cell infectivity). At 18 h after infection, cells were fixed and stained with an anti-N antibody and the % of cells infected was determined by immunofluorescence microscopy. Data are presented as mean values ±SD. $n = 2$ biological replicates. Source data for graphs shown in panels **b–g** are provided as a Source Data file.

cryogenic electron microscopy (cryo-EM) (Supplementary Fig. 5 and Supplementary Table 1). We determined the BriSΔ structure without applying symmetry (C1) at 3.0 Å resolution (Supplementary Fig. 6a). In our analysis, all BriSΔ particles exhibited the locked conformation of the S trimer we had described previously[23]. After applying 3-fold symmetry (C3) we obtained a 2.8 Å cryo-EM map (Fig. 2 and Supplementary Fig. 6b). In this compact locked S conformation, the receptor-binding motif (RBM) is buried inside the RBD trimer obstructing ACE2 receptor binding (Supplementary Fig. 7). - Previously, we

discovered a free fatty acid (FFA) binding pocket in the locked structure of SARS-CoV-2 S, and identified a small molecule tightly bound in the pocket, with the molecular mass of linoleic acid (LA) as determined by electron-spray ionization mass-spectroscopy (ESI-MS)[23], a feature subsequently corroborated in coronavirus S from pangolin[24]. Subsequently, similar density was also identified in other S structures (PDBIDs 6ZB5, 7JJI, 6ZGI, 6ZGE, 6XR8, 6ZP2, and 7DF3. In the locked BriSΔ structure, all three pockets are again occupied by a small molecule (Fig. 2a, b). We chose a method orthogonal to ESI-MS, namely hydrophilic

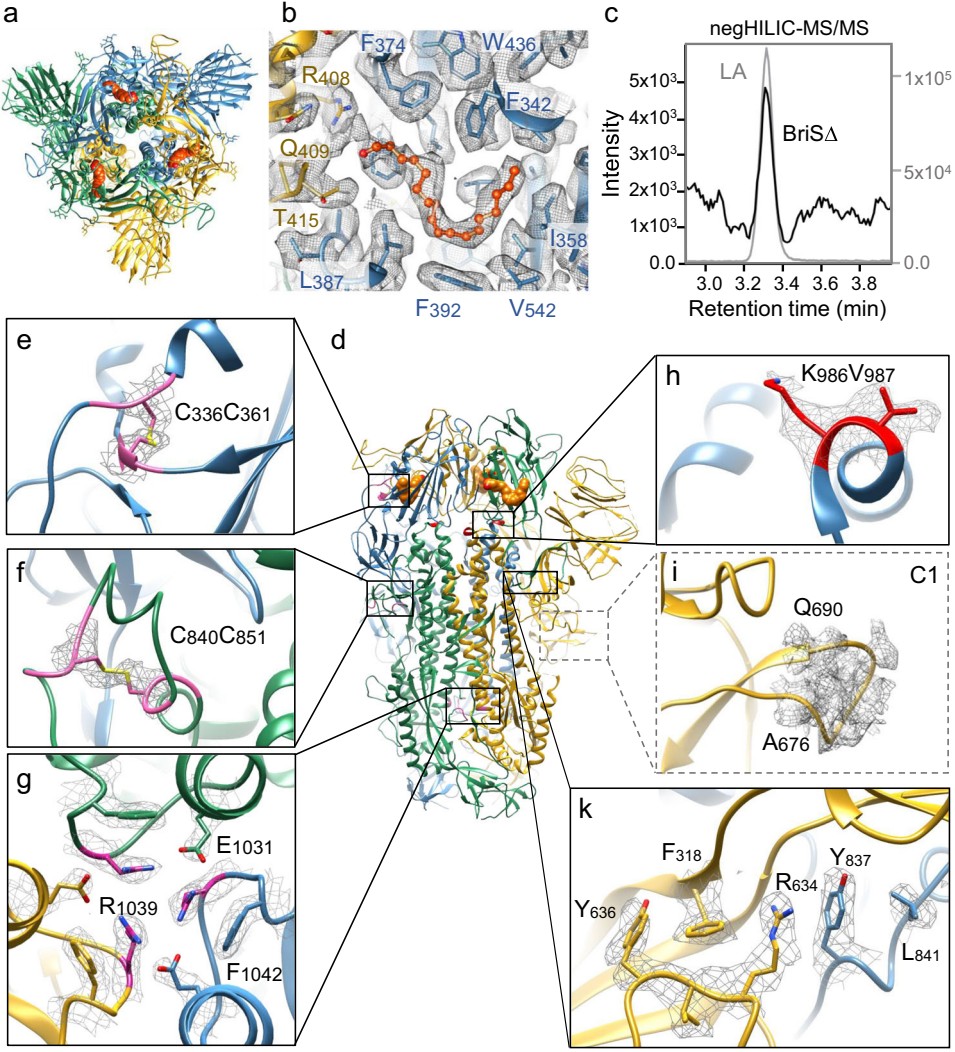

**Fig. 2 Cryo-EM structure of BriSΔ glycoprotein. a** Top view cartoon representation with trimer subunits colored yellow, green and blue, LA shown as orange spheres. **b** Composite LA-binding pocket formed by adjacent RBDs (yellow and blue). EM density is shown as gray-colored mesh; LA ligand (orange) in sticks and balls representation **c** Selected-reaction monitoring mass chromatogram of hydrophilic interaction liquid chromatography (HILIC) coupled tandem mass spectrometry analysis for 10 ng/mL LA analytical standard (gray) and BriSΔ protein preparation (black). Source data are provided as a Source Data file. **d** Side view of BriSΔ trimer with boxes for the close-up views in panels e–k. **e** Disulfide bond between Cys336 and Cys361 in the RBD. **f** Cys840 forms a disulfide bond with Cys851 and stabilizes the fusion peptide proximal region (FPPR). **g** H-bond cluster involving R1039 cation-π interaction on F1042 and forming a salt bridge to E1031. **h** BriSΔ K986 and V987. K986 sidechain EM density indicates flexibility. **i** BriSΔ shortened loop devoid of furin and S1/S2 proteolytic sites modeled as a poly-alanine chain in the C1 structure. **k** R634 cation-π interaction to Y837 in the FPPR of neighboring polypeptide chain.

interaction liquid chromatography followed by tandem mass spectrometry (HILIC-MS-MS) and highly purified LA as a calibration standard, to analyze our BriSΔ glycoprotein samples. Our HILIC-MS-MS analysis provides unambiguous, complementary evidence that the small molecule is indeed LA. In our structure, LA is bound in a bi-partite binding pocket where one RBD provides a hydrophobic 'greasy' tube to accommodate the hydrocarbon tail of LA, while residues R408 and Q409 of the adjacent RBD provide a polar lid coordinating the carboxy head group of LA (Fig. 2b). In the BriSΔ C1 structure, we identified virtually identical tube-shaped densities in all three RBD domains (Supplementary Fig. 7), indicating high occupancy of all three pockets. Using masked 3D classification, we scrutinized the data set for potential heterogeneity in LA binding and found that at least 95% of the RBDs were LA-bound in our structure (Supplementary Fig. 8). Our previous ESI-MS results and the present HILIC-MS-MS results thus are consistent and together identify

the small molecule bound in the FFA-pocket unambiguously as the essential free fatty acid LA.

We scrutinized our BriSΔ structure and compared it with previously determined S structures for conserved stabilizing features (Fig. 2). Disulfide bonds are known to play a crucial role in stabilizing the S trimer and individual domains. Five out of 14 annotated disulfide bonds in S[25] stabilize the RBD including the disulfide bond linking C336 and C361 (Fig. 2d, e). Three arginine R1039 residues, one each from the three polypeptide chains in the S trimer, form a hydrogen bond cluster (Fig. 2g). In this cluster, the arginine residues are symmetrically arranged around the central trimer axis with short-range contacts of 4.65 Å present between the carbon atoms of the guanidino groups. The guanidinium planes stack in a parallel manner on top of the aromatic plane of the juxtaposed F1042, and a salt bridge is formed to E1031 of the adjacent S polypeptide chain (Fig. 2g). R1039, E1031, and F1042 are conserved in all human

coronaviruses, highlighting their central importance. In the vicinity, a disulfide bond is formed by conserved residues C1032 and C1043, arranging E1031 and F1042 at the required distance and in the proper conformation to stabilize the R1039-mediated interaction (not shown). Opening of the RBDs was shown previously to induce an asymmetry in the trimer structure that breaks this H-bond cluster[8,26]. LA binding in the FFA-pockets induces conformational changes to the residues surrounding the FFA-binding pocket in the RBD and beyond, including the NTD, SD2, and the fusion peptide proximal region (FFPR). Re-organization of SD2 in the locked structure results in a stabilization of the region around R634 (Fig. 2k). This arginine residue is stabilized by π-stacking on Y837 and thus connects to the FPPR of the neighboring subunit. Such π-stacking interactions were observed in a previous structure that had an intact furin site and unassigned density in the FFA-binding pocket[10]. R634 stacking to Y837 is additionally stabilized by a hydrophobic interaction (Fig. 2k). Fixed in a rigid position through these interactions, C840 can form a disulfide bond to C851 which additionally stabilizes the FPPR (Fig. 2f). This disulfide bond-mediated stabilization of the FPPR has been described in a cryo-EM structure of full-length S protein comprising the native transmembrane domain[9]. We scrutinized S structures in the protein data bank (PDB) and the sandwiched R634 appears to be a hall mark of the locked conformation. In contrast, S structures in the closed, but not locked, conformation show no π-stacking interaction of R634. Instead, residues 620–640, as well as parts of the FPPR, are disordered, and residues Y636 and R634 adopt different conformation, underscoring a functional link between the locked conformation, the sandwiched R634 and the FFA-binding pocket.

Our BriSΔ construct harbors WT residues K986 and V987, which often are mutated to prolines to stabilize S in a prefusion state. In BriSΔ, the valine fits well into the density while the lysine sidechain appears to be somewhat flexible (Fig. 2h). Importantly, BriSΔ lacks 8 amino acids including the furin cleavage site and the S1/S2 cleavage site located on a flexible loop. This loop is now shorter due to the deletion and thus more rigid, as evidenced by density in the C1 map which allowed to build a poly-alanine chain (Fig. 2i).

*N*-glycosylation of BriSΔ is comparable to previous S structures (Supplementary Table 2). Interestingly, WT residues S673, T676, T678, and S680 close to the furin site are all candidates for O-glycosylation which is dependent on proline P681[27]. It was shown that O-glycosylation of these residues negatively affects furin cleavage, contributing to S stability and infectivity[27]. P681 and S680 are lacking in BriSΔ, and P681 is mutated in other lineages, including the B.1.1.7 variant that emerged in Kent, UK[28] and rapidly spread globally. The enzymes responsible for O-glycosylation (GALNTs) are expressed differently depending on cell type[29], and the absence of P681 in BriSΔ could thus contribute to the observed dominance of this variant in certain cell types. Indeed, it was shown that mutation of P681 to R, which is present in several variants of concerns including the recent rapidly spreading Indian 'delta' variant, increases viral fusion[30].

**Functional analysis of BriSΔ.** The cryo-EM structure of BriSΔ evidenced exclusively particles in the locked conformation in which RBM binding to ACE2 is obstructed (Fig. 2 and Supplementary Figs. 5–7). In cell-based assays, however, BriSΔ virus remains infectious (Fig. 1). To address this apparent discrepancy, we biochemically analyzed the interaction of a range of S proteins with the ACE2 receptor. We compared BriSΔ binding to ACE2 with S protein lacking the RBM, S protein where the furin site

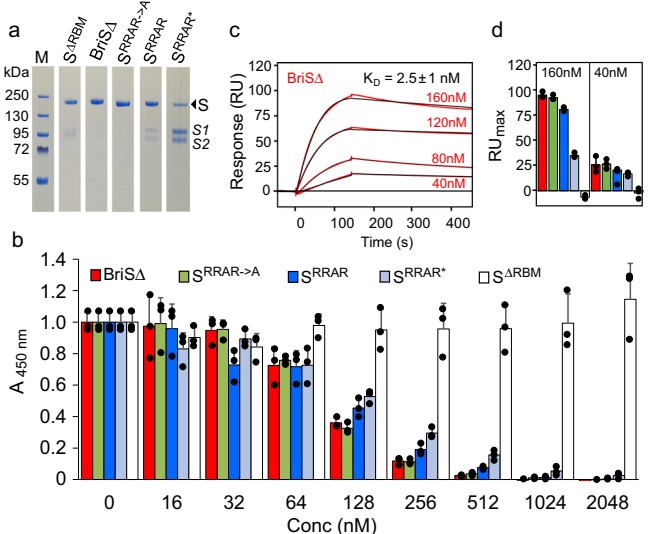

**Fig. 3 Functional analysis of SARS-CoV-2 S proteins. a** Coomassie-stained SDS-PAGE sections of S protein variants used for biochemical characterization. Protein purifications were carried out at least three times each. **b** Competition ELISAs utilizing immobilized ACE2, HRP-labeled RBD, and S proteins (shown in different colors) at the concentrations indicated. Error bars: standard deviations (+SD) are shown, three replicates. **c** Surface plasmon resonance (SPR) of BriSΔ. Concentrations between 40 nM and 160 nM were flowed over 50 RU of biotinylated ACE2 immobilized on a streptavidin-coated sensor chip. Black lines correspond to a global fit of the data using a 1:1 binding model. Each experiment was repeated independently three times. All protein concentrations were used to calculate the $K_D$ value. Source data are provided as a Source Data file. **d** SPR analysis comparing maximal binding (RUmax) of the S protein variants at two representative concentrations (160 nM, 40 nM). Color coding as in panel **b**.

residues are replaced by an alanine (S$^{RRAR \to A}$), uncleaved WT S (S$^{RRAR}$) and furin-cleaved WT S (S$^{RRAR*}$) (Fig. 3a and Supplementary Table 3). ACE2-binding ELISA (performed using a surrogate virus neutralization test kit (sVNT)) indicated that all S proteins efficiently bind ACE2, except the S protein lacking the RBM used as a control (Fig. 3b). For the other S proteins, half-maximal binding was observed between 64 and 128 nM S protein in the assay. We determined the dissociation constant ($K_D$) of BriSΔ and ACE2 by surface plasmon resonance (SPR) with bio-tinylated ACE2 immobilized on a streptavidin-coated chip (Fig. 3c and Supplementary Fig. 9). The binding of BriSΔ ($K_D = 2.5$ nM) to ACE2 is not significantly different as compared to S$^{RRAR \to A}$ (1.4 nM)[23]. In agreement, the maximal RU values, indicating mass deposited on the ACE2-coated chip, were virtually identical for BriSΔ, S$^{RRAR \to A}$ and uncleaved WT S (Fig. 2d and Supplementary Fig. 9a, b). The partially cleaved WT S yielded lower RU$_{max}$ signals, possibly due to partial dissociation of the S trimer and binding of the smaller S1 fragment to ACE2. Our biochemical assays thus establish that the BriSΔ trimer can adopt an open, ACE2-binding competent conformation, which however was not observed on the cryo-EM grids. This finding is in agreement with our results that the BriSΔ variant and WT virus can be neutralized efficiently with similar amounts of a commercial monoclonal antibody recognizing the RBD (Fig. 1g), and with human convalescent sera (Supplementary Fig. 3). Also, similar observations were made in a recent study that identified a fully closed Cryo-EM structure of S with no significant change in ACE2-binding affinity[31].

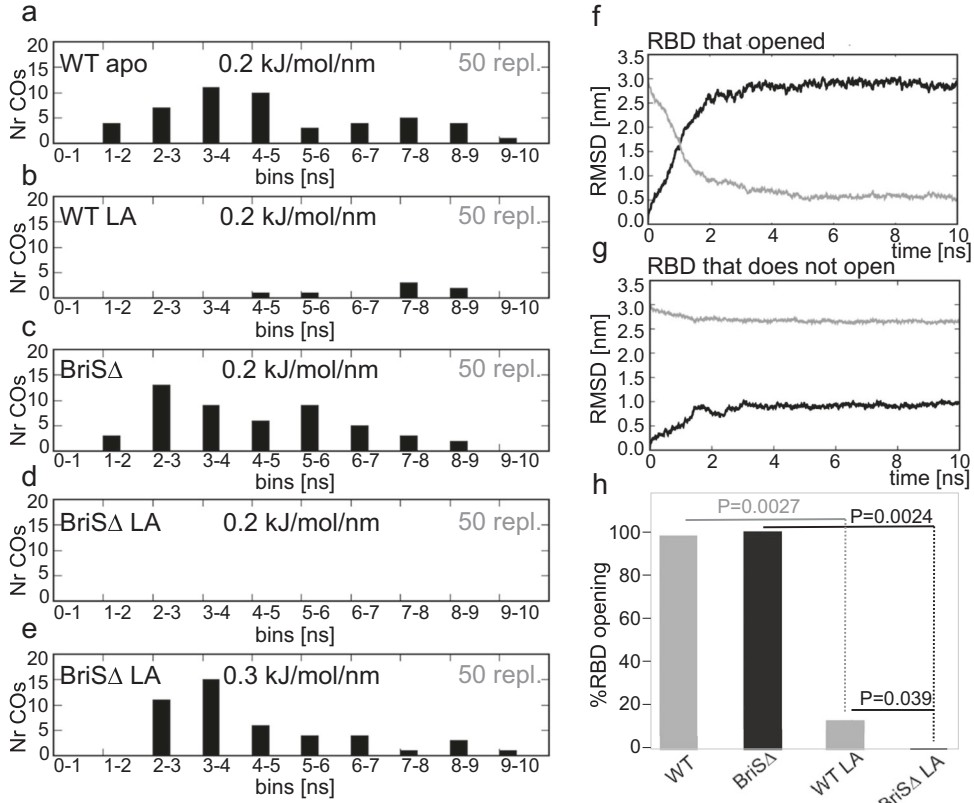

**Fig. 4 Impact of furin-site loop deletion and LA binding on RBD opening. a–e** Distribution of the cross-overs (COs) for 50 × 10 ns replicates carried out for each of the four spike systems under a force constant of 0.2 kJ/mol/nm for **a** WT apo, **b** WT with LA bound **c** BriSΔ apo and **d** BriSΔ with LA bound. **e** Additional force (0.3 kJ/mol/nm) is required to open BriSΔ with LA. **f** Example of a cross-over plot of a system in which the RBD opens. The black trace shows the RMSD from the closed spike (the starting structure). The gray trace shows the RMSD from the open target at time points along the trajectory over which the force was applied. The point at which the traces cross (the cross-over time) is measured in nanoseconds. **g** Example of a cross-over plot in which the RBD fails to cross-over as the RBD does not open under the force constant applied. **h** Comparison of RBD opening events. Significance of the differences between WT apo form (WT-apo) and WT LA-bound form (WT-LA) ($P = 0.002686262$ 99% CI), between BriSΔ apo form (BriSΔ-apo) and BriSΔ LA-bound form (BriSΔ-LA) ($P = 0.002445144$ 99% CI) and WT-LA and BriSΔ-LA ($P = 0.038942552$ 95% CI) was determined by one-tailed, paired T-test with nine degrees of freedom for each data set. Source data for graphs shown in panels **f**, **g** are provided as a Source Data file.

**Targeted molecular dynamics simulations indicate that LA stabilizes locked BriSΔ more than wild-type S.** We next characterized the conformational changes in BriSΔ and WT S by using targeted molecular dynamics simulations (Fig. 4) to explore the force required to open a single RBD from a closed position in the S trimer for apo and LA-bound systems corresponding to WT and BriSΔ. Initially, a range of harmonic force restraints was applied to the WT spike system to find a restraint force under which the RBD opened in half of the test 10 ns simulations. Such conditions allowed an appropriately large number of repeats to be statistically significant over these non-equilibrium simulations. Thus we performed 50 simulations over 10 ns for each system (Fig. 4a–d) applying a harmonic restraint force constant of 0.2 kJ/mol/nm to raise a single RBD from an equilibrated, closed conformation to a target, open state, corresponding to the energy-minimized open S model built on EMD-1146[23]. RMSDs were calculated for C-alpha positions corresponding to the single RBD in each case. Openings in which this RBD moved closer to the open than closed state (i.e. greater than 50% open) were counted as open, as this conformation could still feasibly accommodate an ACE2 receptor interaction[32]. As the same harmonic restraints and RMSD calculation methods were used to compare the WT and BriSΔ systems (apo and LA-bound), the relative differences reflect the degree of stability afforded by either the mutated furin site, the bound LA, or both.

Starting from their closed conformation, we observed that all BriSΔ and almost all apo WT S opened during the 10 ns MD simulations. In contrast, bound LA in the WT S substantially reduces the number of opening events (the cross-over time-point when the RBD opens more than halfway) compared to the apo WT spike (Fig. 4a, b). Notably, the binding of LA to BriSΔ (Fig. 4d) completely prevents RBD opening at 0.2 kJ/mol/nm in these simulations. Increasing the force constant to 0.3 kJ/mol/nm was required to raise the RBD of LA-bound BriSΔ over this time period (Fig. 4e). Example plots illustrate how during the course of the simulation the force applied either resulted in the RBD opening (Fig. 4f and Supplementary Movie 1) or failed to pull the RBD open (no cross-over) as determined by root-mean-square deviation (RMSD) (Fig. 4g) The time at which the RMSD traces cross (cross-over time) occurred was binned (in ns steps) for all 50 replicates for each system, as shown in Fig. 4a–e. In summary, 98% simulations of the apo WT S and 100% of BriSΔ open the RBD with a force constant of 0.2 kJ/mol/nm applied, while only 14% of the WT and 0% of the BriSΔ systems complexed with LA open with the same force applied (Fig. 4h). Thus, LA binding significantly stabilizes the locked conformation of both, WT S ($P = 0.0027$) and BriSΔ ($P = 0.0024$). Importantly, LA apparently shifts the equilibrium favoring the locked conformation for BriSΔ more than for WT S ($P = 0.039$) (Fig. 4h). This hypothesis is supported by our cryo-EM structure which shows that virtually

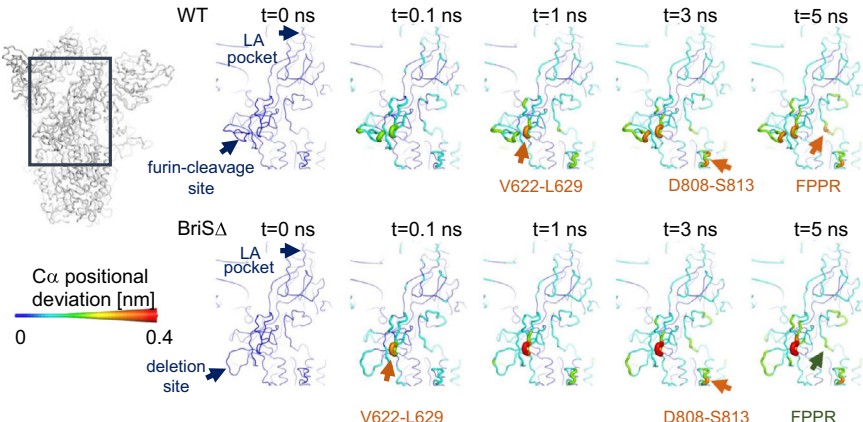

**Fig. 5 Dynamical nonequilibrium simulations elucidate impact of LA on S structure.** Average Cα-positional deviations are shown at 0, 0.1, 1, 3, and 5 ns following LA removal from the FFA-binding pockets of WT S (above) and BriSΔ (below). The Cα deviations between the simulations with and without LA were determined for each residue and averaged over the 90 pairs of simulations for each system. The Cα average deviations are mapped onto the structure used as the starting point for the LA-bound equilibrium simulations. Both colors and cartoon thickness indicate the average Cα-positional deviation values. Important regions of changes are highlighted.

all BriSΔ trimers were LA-bound and in the locked conformation (Fig. 2). The spike protein is glycosylated and though glycosylation probably influences the ease with which the RBD opens, the glycosylation sites of BrisΔ are the same as WT and so the effects exerted by glycosylation are likely to be similar for both cases.

**Communication between the furin cleavage site, the FFA-binding pocket and the FPPR.** We next performed 180 short dynamical-nonequilibrium simulations to analyze intra-spike communication and identify any structural networks connecting the FFA-binding pocket to functionally important motifs within WT S and BriSΔ (Fig. 5 and Supplementary Fig. 10a). In these simulations, the LA molecules were removed from the FFA pockets (from equilibrated simulations, Supplementary Fig 10b) and the response of the systems to this perturbation was determined using the Kubo-Onsager approach[33–35]. The simulations revealed a cascade of structural changes occurring in response to the LA removal and highlight the route by which such changes are transmitted through the S protein (Supplementary Figs. 11 and 12). Strikingly, we find that the furin cleavage site responds swiftly to LA in the WT S, despite being 40 Å away from the FFA pocket (Fig. 5). The nonequilibrium simulations also show allosteric connection between the FFA pocket and V622-L629, D808-S813 and the fusion peptide proximal region (FPPR, residues 833–855) in WT S (Fig. 5 and Supplementary Fig. 11). Similar structural responses are observed for the three WT S subunits (Supplementary Fig. 11). The V622-L629 region is part of a larger loop structure spanning residues 617-641, close to the R634-Y837 cation-π interaction site (Fig. 2k)[36]. The V622-L629 region, close to R634, and the furin-cleavage site respond rapidly to the perturbation in WT S: 0.1 ns after LA removal, conformational rearrangements are observed in both regions (Fig. 5 and Supplementary Fig. 11). As the simulations proceed, a gradual increase in deviations is observed for both the furin-cleavage and V622-L629 regions in WT S, with the conformational changes being further propagated to the segments adjacent to the fusion peptide, namely the FPPR and D808-S813. These results show direct coupling between the FFA-binding pocket and important, distant regions of the protein, including the furin-cleavage site. Our simulations also highlight differences in the dynamic response and in the effect of LA between the WT and BriSΔ spikes (Fig. 5 and Supplementary Figs. 11 and 12): in BriSΔ, only the V622-L629 region responds rapidly to LA

removal. The deletion site itself shows smaller conformational rearrangements in BriSΔ when compared to WT S, confirming a more rigid arrangement of the shortened loop which in WT S comprises the furin cleavage and S1/S2 cleavage sites. These results indicate that BriSΔ has different allosteric and dynamical behavior from the WT S.

**Discussion**

We investigated here the structure and functional characteristics of BriSΔ, a patient-derived SARS-CoV-2 variant we identified as a viral subpopulation by passaging SARS-CoV-2 isolate hCoV-19/ England/02/2020. We find important shared features, and also marked differences between BriSΔ and an artificial ΔPRRA SARS-CoV-2 in which only the four residues of the furin cleavage site had been removed by reverse genetics[19] as well as Sdel, a variant containing a cell passage acquired deletion encompassing the furin cleavage site[18]. On the one hand, all three deletion variants, BriSΔ, ΔPRRA, and SdeI, replicate substantially better in Vero E6 cells but show impaired replication in Calu-3 cells - underscoring that WT SARS-CoV-2 evolved to efficiently infect and replicate in respiratory cells. On the other hand, we find differences highlighting the impact of the exact sequence context of the deletion: expression of serine protease TMPRSS2 in Vero E6 cells reduced the replication advantage of ΔPRRA SARS-CoV-2[19], but not of BriSΔ or Sdel[18]. In our experiments, BriSΔ and WT SARS-CoV-2 both exhibited increased infection and replication in Vero E6/TMPRSS2 cells, but BriSΔ retained higher infection and replication rates as compared to WT virus (Fig. 1c). Importantly, the engineered ΔPRRA virus required more antibodies for neutralization of the virus compared to WT SARS-CoV-2, while monoclonal antibody and human sera neutralize BriSΔ and WT SARS-CoV-2 with equal efficacy (Fig. 1g and Supplementary Fig. 3). Interestingly, ACE2 receptor overexpression in Caco-2 cells improved the infection and replication of BriSΔ and WT SARS-CoV-2 as compared to Caco-2 cells with a lower constitutive expression of ACE2 (Fig. 1d, e).

Mechanistically, the different infectivity of cells expressing high levels of TMPRSS2 is probably explained by the fact that WT SARS-CoV-2 S is cleaved at the furin cleavage site (S1/S2 cleavage), which primes S for cleavage with TMPRSS2 (at S2′) resulting in activation of the fusion peptide[1]. The virus can then fuse at the plasma membrane to enter cells[20,22]. It has been shown that the furin-cleaved S can also interact with neuropilin (NRP1)

following the CendR rule, potentially enhancing viral entry[37,38]. Lacking these modalities, BriSΔ appears to have evolved to be preferentially taken up by receptor-mediated endocytosis and then potentially cleaved by an intracellular protease before fusing via the endosome. Thus, Calu-3 cells which express TMPRSS2 are better infected by WT virus, and they replicate WT virus more efficiently than BriSΔ virus. In addition, intracellular defence pathways exist such as the interferon-induced transmembrane proteins (IFITMs) that may be more active against virus entering through the endosome rather than fusing at the plasma membrane[39]. Vero E6 cells express low to very levels of TMPRSS2, therefore WT virus cannot enter efficiently by fusing at the plasma membrane, and it has recently been suggested that WT SARS-CoV-2 is also taken up by receptor-mediated endo-cytosis in these cells[21]. Our data suggest this process is more efficient for the BriSΔ virus. Notably, Vero E6 cells are deficient in the interferon pathway, and entry via the endosome may not be inhibited in these cells[17]. We hypothesize that deletion of the furin cleavage site potentially results in a more stable viral particle secreted from cells, with increased expression of full-length BriSΔ in a prefusion state, which might result in less infectivity of TMPRSS2 expressing cells compared to WT. BriSΔ, compared to WT virus, showing higher infectivity of Vero E6 cells which have low levels of TMPRSS2 protease and are deficient in the inter-feron pathway also suggests that in the heterogenous environment of the human body such deletion variants can emerge in suitable cell types serving as potential niches for SARS-CoV-2 to further evolve or specialize.

The cryo-EM structure revealed a locked BriSΔ glycoprotein trimer (Fig. 2). Our HILIC-MS-MS results unambiguously iden-tify LA as the small molecule tightly bound in the three bipartite pockets. The LA-bound locked form obstructs both ACE2 bind-ing and integrin-mediated cell entry. We observe two motifs stabilizing LA-bound locked BriSΔ that appear to be character-istic for this S conformation, namely the H-bond cluster formed by R1039 stabilizing the trimer interface (Fig. 2g) and the cation-π interactions between R634 and Y837 stabilizing the interaction of S1 with the S2 of the adjacent trimer subunit (Fig. 2k). Y837 is part of the FPPR motif which is folded in the locked, but not in the closed conformation of S[36]. R634 is part of an extended loop structure that holds the FPPR in place, and alterations in this region, for instance the D614G S mutation, favor the opening of the RBDs and disorder the FPPR[40], possibly exposing the S2′ cleavage site upstream of the fusion peptide[9].

Biochemical interaction assays show that BriSΔ glycoprotein binds ACE2 with comparable affinity as S protein that does not have the deletion (Fig. 3). Thus, BriSΔ can adopt open con-formation(s), consistent with the infectivity of the BriSΔ variant (Fig. 1). We conclude that cryo-EM analysis, while indicating clear preference for the locked form by the BriSΔ trimer, may not fully reflect the dynamics of the S trimer which is subject to a range of stabilizing and destabilizing modalities[41]. A decreased stability of open BriSΔ trimer likely derives from the more rigid loop (T676-Q690) lacking 8 amino acids including two protease cleavage sites. In agreement with this, it has been suggested that furin cleavage facilitates the opening of WT S[10] and the presence of ACE2 receptors enhances the opening of the RBDs[42]. Our study indicates that the locked BriSΔ conformation is stabilized as compared to WT S by the additional interactions we described within the glycoprotein trimer, and the shortened loop may assist to keep these properly in place. Thus, BriSΔ may have evolved different opening kinetics from WT S and other variants, without noticeably affecting the equilibrium binding constant to ACE2. Consistent with this, our molecular dynamics simulations clearly support different stability and kinetics of RBD opening of LA-bound BriSΔ and WT S. In our targeted MD simulations,

LA-bound BriSΔ requires significantly more exertion of force as compared to WT S, to adopt an open conformation (Fig. 4). Nonequilibrium MD simulations show that in WT S, both the furin site and the V622-L629 region respond rapidly to LA removal; after 0.1 ns conformational changes are observed in both regions (Fig. 5). In contrast, in BriSΔ, a fast conformational response is only observed for the V622-L629 region, indicating effectively no communication between the shortened loop and the LA pocket (Fig. 5). The nonequilibrium simulations confirm allosteric connections between the LA pocket and regions V622-L629, D808-S813 and the FPPR (Fig. 5), which are stabilized by R634 π-stacking interaction with Y837 in the FPPR in the locked conformation (Fig. 2k). LA removal destabilizes these regions.

Appreciation of the scale of intra-patient sequence variability in infection by HIV – likewise an RNA virus - led to a much better understanding of the HIV infection lifecycle and the rea-lization of the need for combination drug therapy[43]. Similarly, evidence accumulates for intra-host genetic diversity in SARS-CoV-2[44,45]. The identification of viruses with furin cleavage site deletions in clinical samples from COVID-19 patients[16,17] and the differential infectivity and replication of the WT and BriSΔ viruses suggests that in the heterogenous environment of the human body, SARS-CoV-2 variants with abrogated S1/S2 clea-vage replicate preferentially in specific cell types. While WT SARS-CoV-2 clearly has a competitive advantage in respiratory tract cells[19], the BriSΔ virus could preferentially infect and replicate in other human cell types as a niche and reservoir to delay full clearance of SARS-CoV-2 infection by the immune system. Indeed, post-mortem tissue analysis revealed viruses with a furin cleavage site deletion in heart and spleen tissue which could represent such reservoirs[17]. Previous studies of RNA viruses have empirically demonstrated that the phenotype of a viral population can change measurably without changes in the consensus sequence due to increased diversity in evolved virus subpopulations[46]. We propose that BriSΔ exemplifies such SARS-CoV-2 evolution, with variants of the virus exploiting different cells and tissue types as niches for probing evolutionary space resulting in intra-host genetic and functional diversity, under-scoring the need for analyzing individual virus genomes rather than viral populations for better understanding of transmission chains and pathology, and development of future treatments to overcome SARS-CoV-2.

## Methods

### SARS-CoV-2 propagation and assay

*Cell culture*. Human Calu-3 (ATCC® HTB-55™), Caco-2 (ATCC® HTB-37™; a kind gift from Dr Darryl Hill) and African green monkey Vero E6 (ATCC® CRL 1586™) cell lines were obtained from the American Type Culture Collection. A Caco-2 cell line expressing ACE2 (Caco-2-ACE2; a kind gift from Dr Yohei Yamauchi, Uni-versity of Bristol) and Vero E6 cells modified to constitutively express TMPRSS2 (Vero E6/TMPRSS2 cells[5]; obtained from NIBSC, UK) were also used in the study. Cells were cultured in Dulbecco's modified Eagle's medium plus GlutaMAX (DMEM, Gibco™, ThermoFisher) supplemented with 10% fetal bovine serum (FBS) and 0.1 mM non-essential amino acids (NEAA, Sigma Aldrich) except Calu-3 cells, that were grown in Eagle's minimal essential medium plus GlutaMAX (MEM, GibcoTM, ThermoFisher) supplemented with 10% FBS, 0.1 M NEAA, and 1 mM sodium pyruvate. All cells were grown at 37 °C in 5% $CO_2$.

*SARS-CoV-2 isolation and sequencing*. A mixed virus population containing the SARS-CoV-2 WT isolate hCoV-19/England/02/2020 (GISAID ID: EPI_ISL_407073) and the "Bristol" variant derived from it (BriSΔ) in which spike amino acids 679-687 (NSPRRARSV) had been deleted and replaced with Ile[11] was grown on either Vero E6 cells or Caco-2 cells and a single virus population was isolated after two rounds of limiting dilution in either cell line (Supplementary Fig. 1). In brief, the virus samples were serially diluted and grown on either Vero E6 or Caco-2 cells in order to favor the growth of the BriSΔ variant or the WT virus respectively. After 5 days of incubation, the culture supernatants in wells showing cytopathic effect (CPE) at the highest dilution were again diluted and the process was repeated. An aliquot of culture supernatant from wells showing CPE at the highest dilution was used for RNA extraction and RT-PCR using a primer set

designed to discriminate the wild-type and BriSΔ viruses. Stocks of the purified viruses were produced and quantified for genome copy number by qRT-PCR and titered on Vero E6 and Caco-2/ACE2 cells as described previously[11,23]. To verify the viral genome sequences, the wildtype and BriSΔ variant virus were grown for 24 h in Vero E6 cells before harvesting the cells and extracting the total RNA with Trizol reagent (ThermoFisher) prior to direct RNA sequencing using an Oxford Nanopore flow cell as previously described[47]. The sequenced reads were aligned to the SARS-CoV-2 genome using minimap2 and the S gene was assessed visually for deletions as well as by an in-house script designed to look for significant deletions in this region and across the whole genome. All work with infectious SARS-CoV-2 was done inside a class III microbiological safety cabinet in a containment level 3 facility at the University of Bristol.

*SARS-CoV-2 infection and growth assays.* Cells were seeded the day prior to infection in appropriate media in μClear 96-well Microplates (Greiner Bio-one). The culture supernatants were removed, and cells were infected with WT or BriSΔ SARS-CoV-2 in infection medium (MEM with GlutaMAX supplemented with 2% FBS and 0.1 M NEAA). For infection assays, virus was diluted in an 8-step 2-fold dilution series from neat virus (equal genome copy numbers) in triplicate. For growth assays, cells were infected with MOI 0.5 based on cell type-dependent titers from Caco-2/ACE2 and Vero E6 cells; one plate was infected per time point containing 6 replicates per condition. After 1 h of infection at room temperature, the virus was removed and replaced with an infection medium and incubated at 37 °C in 5% CO$_2$. At assay-dependent times (18 h for infection assay and 24/48/ 72 hours for growth assays) the cells were fixed in 4% (v/v) paraformaldehyde for 60 minutes for image analysis. Fixed cells were permeabilized with 0.1% Triton-X100 in PBS and blocked with 1% (w/v) bovine serum albumin before staining with a monoclonal antibody against the SARS-CoV-2 nucleocapsid protein (N) (1:2000 dilution; 200-401-A50, Rockland) followed by an appropriate Alexa Fluor-conjugated secondary antibody (1:2000 dilution; ThermoFisher-11008 and life technologies-11036) and DAPI (Sigma Aldrich). To determine the number of virus-infected cells, images were acquired on an ImageXpress Pico Automated Cell Imaging System (Molecular Devices) using the 10X objective. Stitched images of 9 fields covering the central 50% of the well were analyzed for virus-infected cells using Cell Reporter Xpress software (Molecular Devices). The cell number was determined by automated counted of DAPI stained nuclei and infected cells were identified as those cells in which positive N staining was detected associated with nuclear DNA staining. For growth assays, supernatants from the six replicate wells were collected and pooled prior to cell fixation, and viral RNA content was quantified by qRT-PCR as previously described[23]. Statistical significance was assessed using an unpaired Student's test in GraphPad Prism v8.4.3.

*SARS-CoV-2 neutralization assays.* For virus neutralization assays Vero E6/ TMPRSS2 cells were seeded the day prior to infection in appropriate media in μClear 96-well Microplates. Before infection, a commercial monoclonal antibody (Absolute Antibody; Sb#15) recognizing the S protein receptor-binding domain (RBD) or, alternatively, heat-inactivated (30 min at 56 °C) convalescent serum from a SARS-CoV-2 infected individual, were serially diluted, in duplicate, in infection medium over an 8-fold dilution range. Equal amounts of the wild type and BriSΔ viruses (based on Vero E6 cell infectivity) diluted in infection medium, were mixed with the antibody/antisera dilutions and incubated for 60 minutes at 37 °C. Following the incubation, the culture supernatants were removed from the cells and replaced with the virus: antibody/serum dilutions followed by incubation for 18 h at 37 °C in 5% CO$_2$. Cells were then fixed, and the number of infected cells was determined by immunofluorescence assay and image analysis as described above. The percentage of infected cells relative to control wells infected with virus and infection media only were calculated and variable slope non-linear fit curves were assigned using GraphPad Prism v8.4.3.

## Protein expression and purification

*BriSΔ protein.* The construct encoding BriSΔ was synthesized at Genscript (Genscript Inc, New Jersey USA) followed by cloning into pACEBac1 plasmid (Geneva Biotech, Switzerland). The expression construct comprises SARS-CoV-2 spike amino acids 1 to 1208 with the 8 amino acid deletion, followed by a linker, a T4-foldon trimerization domain, another linker and finally an octahistidine affinity purification tag (Supplementary Table 3). BriSΔ was produced with the MultiBac baculovirus expression system (Geneva Biotech)[48] in Hi5 cells using ESF921 media (Expression Systems Inc.). Supernatants from transfected cells were harvested 3 days post-transfection by centrifugation of the culture at 1000×*g* for 10 min followed by another centrifugation of supernatant at 5000×*g* for 30 min. The final supernatant was incubated with 10 mL HisPur Ni-NTA Superflow Agarose (Thermo Fisher Scientific) per 3 L of culture for 1 h at 4 °C. Subsequently, a gravity-flow column was used to collect the resin bound with BriSΔ, followed by washing with 30 column volumes (CV) of wash buffer (65 mM NaH$_2$PO$_4$, 300 mM NaCl, 20 mM imidazole, pH 7.5), 30 CV high salt buffer (65 mM NaH$_2$PO$_4$, 1000 mM NaCl, 20 mM imidazole, pH 7.5), and again 30 CV wash buffer. BriSΔ protein was eluted using a step gradient of elution buffer (65 mM NaH$_2$PO$_4$, 300 mM NaCl, 235 mM imidazole, pH 7.5). Elution fractions were analyzed by reducing Coomassie-stained SDS-PAGE. Fractions containing BriSΔ were pooled, concentrated using 50 kDa MWCO Amicon centrifugal filter units (EMD Millipore)

and buffer-exchanged in SEC buffer (20 mM Tris, pH 7.5, 100 mM NaCl). Concentrated BriSΔ was subjected to size exclusion chromatography (SEC) using a Superdex 200 increase 10/300 column (GE Healthcare) equilibrated in SEC buffer. Peak fractions were analyzed by reducing SDS-PAGE and negative stain electron microscopy (EM) (Supplementary Fig. 4); fraction 8 was used for cryo-EM.

*S$^{ΔRBM}$ protein.* The construct encoding S$^{ΔRBM}$ was synthesized by Genscript (New Jersey USA) and subcloned into pACEBac1 plasmid (Geneva Biotech). S$^{ΔRBM}$ comprises, as described above for BriSΔ, a T4-foldon trimerization domain and an octahistidine tag. In S$^{ΔRBM}$ the ACE2-interacting receptor-binding motif (RBM) in the RBD is deleted and replaced with a glycine-serine rich linker. In addition, S$^{ΔRBM}$ comprises mutations K986P and V987P (Supplementary Table 3). Protein was produced and purified as described above except that the SEC was performed in 1x PBS at pH 7.5.

*S$^{RRAR}$ protein.* The construct encoding S$^{RRAR}$ was synthesized and cloned into pACEBac1 (Genscript). S$^{RRAR}$ comprises an intact furin site, mutations K986P and V987P, the T4 foldon trimerization motif and an octahistidine affinity purification tag (Supplementary Table 3). S$^{RRAR}$ was produced and purified as described above for S$^{ΔRBM}$.

*S$^{RRAR*}$ protein.* S$^{RRAR*}$ was obtained by incubating S$^{RRAR}$ with furin protease (New England Biolabs) supplemented with 1 mM CaCl$_2$ and incubated at room temperature for 8 hours.

*S$^{RRAR->A}$ protein.* S$^{RRAR->A}$ protein was produced and purified as described[23]. In S$^{RRAR->A}$, the furin cleavage site RRAR is replaced with a single alanine residue[49] (Supplementary Table 3).

**Negative stain sample preparation and microscopy**. In all, 4 μL of 0.05 mg/mL SARS-CoV-2 spike protein was applied onto a freshly glow discharged (1 min at 10 mA) CF300-Cu-50 grid (Electron Microscopy Sciences), incubated for 1 min, and manually blotted. In total, 4 μL of 3% uranyl acetate was applied onto the same grid and incubated for 1 min before the solution was blotted off. The grid was loaded onto a FEI Tecnai12 120 kV BioTwin Spirit TEM. Images were acquired at a nominal magnification of ×49,000.

**Cryo-EM sample preparation and data collection**. In total, 4 μL of 1.25 mg/mL BriSΔ was loaded onto a freshly glow discharged (2 min at 4 mA) C-flat R1.2/1.3 carbon grid (Agar Scientific), blotted using a Vitrobot MarkIV (Thermo Fisher Scientific) at 100% humidity and 4 °C for 2 s, and plunge frozen. Data were acquired on a FEI Talos Arctica transmission electron microscope operated at 200 kV and equipped with a Gatan K2 Summit direct detector and Gatan Quantum GIF energy filter, operated in zero-loss mode with a slit width of 20 eV using the EPU software. Data were collected in super-resolution at a nominal magnification of ×130,000 with a virtual pixel size of 0.525 Å. The dose rate was adjusted to 6.1 counts/ physical pixel/s. Each movie was fractionated in 55 frames of 200 ms. In all, 8639 micrographs were collected in a single session with a defocus range comprised between −0.8 and −2 μm.

**Cryo-EM data processing**. The dose-fractionated movies were gain-normalized, aligned, and dose-weighted using MotionCor2[50]. Defocus values were estimated and corrected using the Gctf program[51]. 1,168,229 particles were automatically picked using Relion 3.0 software[52]. The auto-picked particles were extracted with a box size of 110 px (2x binning). Reference-free 2D classification was performed to select well-defined particles. After four rounds of 2D classification, a total of 403,252 particles were selected for subsequent 3D classification. The initial 3D model[23] was filtered to 60 Å during 3D classification in Relion using 8 classes. 199,758 particles from Class 6 (Supplementary Fig. 5) were re-extracted with a box size of 220 px (1.05 Å/px, unbinned) and used for subsequent 3D refinement. The 3D-refined particles were then subjected to a second round of 3D classification using 3 classes. Class 2 and 3 were combined yielding 196,832 particles. These particles were subjected to 3D refinement without applying any symmetry. The maps were subsequently subjected to local defocus correction and Bayesian particle polishing in Relion 3.1. Global resolution and B factor (−97.6 Å²) of the map were estimated by applying a soft mask around the protein density, using the gold-standard Fourier shell correlation (FSC) = 0.143 criterion, resulting in an overall resolution of 3.03 Å. C3 symmetry was applied to the Bayesian polished C1 map using Relion 3.1, with 590,496 particles yielding a final resolution of 2.80 Å (B factor of -106.7 Å²). Local resolution maps were generated using Relion 3.1 (Supplementary Fig. 6).

**Cryo-EM model building and analysis**. For model building, UCSF Chimera[53] was used to fit an atomic model of the SARS-CoV2 Spike locked conformation (PDB ID 6ZB5[23]) into the C3-symmetrized BriSΔ cryo-EM map. The model was rebuilt using sharpened[54] and unsharpened maps in Coot[55] and then fitted into the C1 cryo-EM map. Namdinator[56] and Coot were used to improve the fit and N-linked glycans were built into the density for both models where visible. Restraints for non-standard

ligands were generated with eLBOW[57]. The model for C1 and C3-symmetrized closed conformation was real space refined with Phenix[58], and the quality was additionally analyzed using MolProbity[59] and EMRinger[60], to validate the stereo-chemistry of the components. Figures were prepared using UCSF chimera and PyMOL (Schrodinger, Inc).

**Masked 3D classification in Relion 3.1.** The refined C3 particles stack was expanded 3-fold according to C3 symmetry in Relion. The symmetry-expanded particle stack was then used as input for the masked 3D classification with the focus mask corresponding to individual single chain subunits. Masked 3D classification[61] was performed without alignment using 5 classes for C3-expanded particles (Supplementary Fig. 8). Visual inspection of the 3D classes showed 95% of the chains clustered in the first 3 classes with LA-bound in the hydrophobic pocket within the RBD.

**Mass spectrometry.** LA detection was performed by multi reaction monitoring (MRM) experiments using a Sciex QTrap 4500 system coupled to hydrophilic interaction liquid chromatography. The Analyst 1.7.0 software from Sciex was used for instrument control. For calibration, an unprocessed LA (Sigma Aldrich, Germany) sample dissolved in acetonitrile was used. For spike sample preparation, 100 μL of purified protein sample (1.09 mg/mL) was mixed with 400 μL chloroform for 2 h on a horizontal shaker in a teflon-sealed glass vial at 25 °C. Subsequently, the top organic phase was transferred to a new glass vial and placed in a desiccator for evaporation of the chloroform for 30 min. After all chloroform was evaporated, 50 μL acetonitrile was added to dissolve the fatty acids. From this solution, 10 μL were injected for MRM experiments. A flow rate of 0.55 mL/min was used for the binary flow elution program with acetonitrile (solvent B). The measurements were performed in negative ionization mode. Source ionization, precursor selection and fragmentation parameters for LA monitoring using hydrophilic interaction liquid chromatography were: curtain gas = 35 psi; temperature = 600 °C; nebulizer gas = 65 psi; heater gas = 80 psi; collision gas = 9; ionization voltage = −4500V; Q1 = 279.2 m/z; Q3 = 261.2; dwell time = 250 ms; collision energy = −24 V; declustering voltage = −100 V; cell exit potential = −11 V; entrance potential = −10 V. The recorded data were analyzed using the MultiQuant 3.0.2 software from Sciex.

**Surrogate virus neutralization test assay.** The SARS-CoV-2 surrogate virus neutralization test (sVNT) kit was obtained from GenScript Inc. (New Jersey, USA). Serial dilution series were prepared (from 0-2048 nM) of the different purified SARS-CoV-2 S proteins and RBD in PBS pH 7.5. This dilution series was treated as described[23]. Briefly, dilution series were mixed with the same volume of diluted horseradish peroxidase-conjugated RBD (HRP-RBD) from the sVNT kit and incubated at 37 °C for 30 min. In all, 100 μL of the mixtures were then added to the ELISA plate wells coated with ACE2, according to the manufacturer's protocol. Triplicates of each sample were made. The plate was then incubated at 37 °C for 15 min sealed with tape to avoid evaporation and subsequently washed 4 times with 1x wash buffer. The signal was developed by adding tetramethyl-benzidine (TMB) solution to each well, incubating 15 min in the dark, followed by adding stop solution supplied with the kit. The absorbance at 450 nm was immediately recorded. The data was plotted using Microsoft excel. Standard deviations from three independent replicates were added as error bars.

**Surface plasmon resonance experiments.** Interaction experiments using surface plasmon resonance (SPR) were carried out with a Biacore T200 system (GE Healthcare) according to the manufacturer's protocols and recommendations. Experiments were setup as described before[23]. Briefly, purified biotinylated ACE2[23] was immobilized on a streptavidin-coated (SA) chip (GE Healthcare) at ~50 RUs. BriSΔ was injected at concentrations of 40 nM, 80 nM,120 nM, and 160 nM. S[ΔRBM], S[RRAR->A], S[RRAR], and S[RRAR*] were injected at 40 nM and 160 nM. The running buffer for all measurements was PBS buffer pH 7.5. Sensorgrams were analyzed and K$_D$, k$_{on}$ and k$_{off}$ values were determined with the Biacore Evaluation Software (GE Healthcare), fitting the raw data using a 1:1 binding model. All experiments were performed in triplicates.

**Molecular dynamics simulations.** The starting structures were comprised of closed WT (WT, P0DTC2) and furin-site deletion (BriSΔ) spike trimers, which had either been equilibrated in the presence of linoleic acid (LA) or not (apo). The target structure was the corresponding S trimer in which a single chain had an RBD in an extended conformation as in EMD-11146[23]. Exploratory forces were applied to all four systems to find a value that resulted in a distribution of openings when subjected to 10 ns dynamic simulation. Hence starting structures were simulated for 10 ns (each system was replicated 50 times) with harmonic restraints of 0.2 kJ/mol/nm force applied to pull the RBD C-alphas to align with those of the corresponding C-alphas of the RBD of the open structure corresponding to (EMD-11146[23]. The BriSΔ LA-bound system was subjected to a further 50 replicate simulations at k = 0.3 kJ/mol/nm, because it failed to open at k = 0.2 kJ/mol/nm. A paired T-test was applied to test for significance when comparing the data sets.

Protein coordinates were prepared as described[23]. Loops for the unstructured regions of the locked (LA-bound) and open (apo) cryo-EM structures were built using Chimera (UCSF)[62]. Loop deletions were carried out in Chimera for the furin-site deletion (BriSΔ) and the sequences were verified by Clustal[63] alignment with the published sequence[11]. Likely disulfide bonds were reconstructed based on experimentally observed distances and each chain observe was used in an EBI-blast check to verify WT Spike sequence post build. PROCHECK[64] was then used to check the quality of the resulting structure prior to simulation. ACPYPE[65] was used to prepare the topologies for linoleic acid. For the purposes of comparison for the targeted MD the first 10 residues of the open chain were removed so that both the starting and open target structures had the same number of residues.

*Simulation details.* All simulations were performed under the Amber99SB-ildn[66–68] forcefield in NPT ensembles at 310 K using periodic boundary conditions. We have previously shown that these protocols give structures in good agreement with experiment for the WT spike. Hydrogen atoms, consistent with pH 7, were added to the complex. Short-range electrostatic and van der Waals interactions were truncated at 1.4 nm while long-range electrostatics were treated with the particle-mesh Ewald method and a long-range dispersion correction was applied. A simulation box extending 2 nm from the protein was filled with TIP3P water molecules and 150 mM Na$^+$ and Cl$^−$ ions were added to attain a neutral charge overall. The pressure was controlled by the Berendsen barostat and temperature by the V-rescale thermostat. The simulations were integrated with a leapfrog algo-rithm over a 2 fs time step, constraining bond vibrations with the P-LINCS method. Structures were saved every 0.1 ns for analysis in each of the simulations over 100 ns. Simulations were run on the Bristol supercomputer BlueCrystal4, the BrisSynBio BlueGem, and the UK supercomputer, ARCHER.

*Software.* The GROMACS-2019.2[69] suite of software was used to set up and per-form the molecular dynamics simulations and analyses. Molecular graphics manipulations and visualizations were performed using VMD-1.9.1[70] and Chimera-1.10.2[62].

**Nonequilibrium simulations.** To map the structural changes associated with the LA removal from the FFA-binding pocket in the WT and BriSΔ spike proteins, two sets of 90 dynamical-nonequilibrium simulations were performed. In this approach, the response of a system to a perturbation is directly computed by calculating the difference of a given property between simulations with and without a perturbation. Subtracting the perturbed and unperturbed pairs of simulations at a given time, and averaging the results over many replicates, allows for the identi-fication of the events associated with signal propagation and the determination the statistical significance of the observation[33–35,71]. The perturbation was generated by the (instantaneous) removal of LA from the FFA-binding pocket. Note that the perturbation used here is not intended to represent the physical process of unbinding, but rather is designed to force the system out of equilibrium and drive a rapid response within the protein, as it adapts to LA removal. A graphical repre-sentation of the procedure is shown in Supplementary Fig. 10a. Three equilibrium MD simulations, 200 ns each, were performed for the locked form of the ungly-cosylated head region of the WT and BriSΔ S proteins with LA bound. All equi-librium simulations were considered sufficiently equilibrated after 50 ns (Supplementary Fig. 10b). The simulation conditions for the equilibrium and dynamical-nonequilibrium simulations were identical to those described in[23,72]. The starting conformations for the nonequilibrium simulations were obtained from the equilibrated part of the equilibrium LA-bound simulations. Conformations were taken every 5 ns, and each of the LA molecules were (instantaneously) removed from all the FFA-binding pockets. The resulting apo system was then simulated for 5 ns (Supplementary Figs. 11 and 12). 180 apo dynamical-nonequilibrium simulations were performed (90 simulations each for BriSΔ and WT S). Three positive ions were also removed from the solvent to maintain the electroneutrality of the systems (required for the PME method[73]).

The Kubo-Onsager approach[33–35,71] was used to extract and characterize the structural changes in the protein associated with LA removal. The response to perturbation is directly measured by averaging a given property (in this case, the position of the Cα atoms) in the perturbed (apo) and unperturbed (LA-bound) simulations at a given time, for multiple pairs of trajectories (Supplementary Fig. 10a). For each pair of LA-bound equilibrium and perturbed apo dynamical-nonequilibrium simulations, the difference in positions of each Cα was determined at equivalent points in time, namely after 0, 0.1, 1, 3, and 5 ns of simulation. The pairwise comparison between the positions of Cα atoms allows for the direct identification of the conformational rearrangements while reducing the noise coming from side-chains fluctuation. The Cα-positional deviations at each point in time were averaged over all 90 replicates. The statistical significance of the structural changes identified here is demonstrated by a low standard error of the averages (data not shown). Upon LA removal, the FFA pocket contracts becoming occupied by the sidechain of the residues lining the pocket (Supplementary Figs. 13–15). Additionally, no breaking of the cation-π interaction between R634 and Y837 (Supplementary Fig. 16) or unfolding of the Y837 region was observed in BriSΔ upon LA removal. Similarly, no significant changes were observed in the network of interactions formed by R1039 in the 5 ns following LA removal (Supplementary Figs. 17 and 18).

**Statistics**. Statistical significance was determined by calculating standard deviations following standard mathematical formulae. Standard deviations were calculated from independent triplicates unless indicated otherwise.

**Reporting summary**. Further information on research design is available in the Nature Research Reporting Summary linked to this article.

## Data availability

The data that support this study are available from the corresponding authors upon reasonable request. Structural datasets and coordinates generated during the current study have been deposited in the Electron Microscopy Data Bank (EMDB) under accession numbers EMD-12818 (C3 structure) and EMD-12842 (C1 structure) and in the Protein Data Bank (PDB) under accession numbers: 7OD3 (C3 structure) and 7ODL (C1 structure). Reagents are available from K.G., C.S. and I.B. under a material transfer agreement with the University of Bristol. Source data are provided with this paper.

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

## Acknowledgements
We thank all members of the Berger and Schaffitzel teams as well as Robin Shattock (Imperial College, UK) and Adam Finn (Bristol UNCOVER Group and Children's Vaccine Centre, Bristol Medical School) for their assistance and advice. We thank Simon Burbidge, Thomas Batstone and Matt Williams for computation infrastructure support. We would like to thank the Advanced Computing Research Centre (ACRC) at the University of Bristol for access to BlueCryo, BlueCrystal Phase 4 and BlueGEM, and the UK HECBioSim for access to the UK supercomputer, ARCHER. We are particularly grateful to Thiru Thangarajah (Genscript) for early access to Genscript's cPass™ SARS-CoV-2 Neutralization Antibody Detection/Surrogate Virus Neutralization Test Kit (L00847). We thank Sebastian Fabritz and the Core Facility for Mass Spectrometry at the Max Planck Institute for Medical Research for their support on MS measurements. For the purpose of Open Access, the authors have applied a CC BY public copyright license to any Author Accepted Manuscript version arising from this submission. This research received support from the Elizabeth Blackwell Institute for Health Research and the EPSRC Impact Acceleration Account EP/R511663/1, University of Bristol, from Bris-SynBio a BBSRC/EPSRC Research Centre for synthetic biology at the University of Bristol (BB/L01386X/1) (I.B., C.S., A.J.M., D.K.S., and A.S.F.O.) and from the BBSRC (BB/P000940/1) (C.S. and I.B.). This work received generous support from the Oracle Higher Education and Research program to enable cryo-EM data processing using Oracle's high-performance public cloud infrastructure (https://cloud.oracle.com/en_US/cloud-infrastructure) and the EPSRC through a COVID-19 project award via HECBioSim to access ARCHER (A.J.M.). We acknowledge support and assistance by the Wolfson Bioimaging Facility and the GW4 Facility for High-Resolution Electron Cryo-Microscopy funded by the Wellcome Trust (202904/Z/16/Z and 206181/Z/17/Z) and BBSRC (BB/R000484/1). The authors are grateful to University of Bristol's Alumni and Friends, which funded the ImageXpress Pico Imaging System. O.S. acknowledges support from the Elisabeth Muerer Foundation, the Max Planck School Matter to Life and the Heidelberg Biosciences International Graduate School. J.S. is the Weston Visiting Professor at the Weizmann Institute of Science, part of the excellence cluster CellNetworks at Heidelberg University and acknowledges funding from the European Research Council (ERC, contract no. 294852), SynAd and the MaxSynBio Consortium, funded by the Federal Ministry of Education and Research of Germany and the Max Planck Society, from the SFB 1129 and Project 240245660-SFB1129 P15 of the German Research Foundation (DFG) and from the Volkswagen Stiftung (priority call "Life?"). A.D.D. and D.A.M. are supported by the United States Food and Drug Administration (HHSF223201510104C) and UK Research and Innovation/Medical Research Council (MRC) (MR/V027506/1). M.K.W is supported by MRC grants MR/R020566/1 and MR/V027506/1 (awarded to A.D.D). A.J.M. is supported by the British Society for Antimicrobial Chemotherapy (BSAC-COVID-30) and the EPSRC (EP/M022609/1, CCP-BioSim). I.B. acknowledges support from the EPSRC Innovative Future Vaccine Manufacturing and Research Hub (EP/R013764/1). C.S. and I.B. are Investigators of the Wellcome Trust (210701/Z/18/Z; 106115/Z/14/Z).

## Author contributions
C.S. and I.B. conceived and guided the study. K.G. and F.G. produced proteins, K.G. purified and analyzed all protein samples and carried out biochemical and biophysical experiments, S.K.N.Y. and U.B. prepared grids and collected EM data, S.K.N.Y. carried out image processing, C.T. carried out model building and structural analysis. A.D.D. and M.K.W. performed all live virus CL3 work and analyzed data. D.A.M and A.A. performed direct RNA sequencing and analysis of sequence data. D.K.S., A.S.F.O., and A.J.M. performed all MD simulations. O.S. and J.S. performed and interpreted mass spectrometry. I.B., D.F., and C.S. wrote the manuscript with input from all authors.

## Competing interests
F.G. and I.B. report shareholding in Imophoron Ltd, unrelated to this Correspondence. D.F. and I.B. report shareholding in Geneva Biotech SARL, unrelated to this Correspondence. C.S., D.F., and I.B. report shareholding in Halo Therapeutics Ltd related to this Correspondence. Patent applications describing methods, material compositions and formulations based on the present observations have been filed. The remaining authors declare no competing interests.
