## [Peer Review File · Nature Communications]

Structural insights in cell-type specific evolution of intra-host diversity by SARS-CoV-2Reviewers' Comments:

Reviewer #1:

Remarks to the Author:

In this manuscript the authors used a variety of experimental and MD simulation techniques to understand how deletion of 8 amino acid residues in the furin recognition motif and S1/S2 cleavage site of SARS-Cov-2, a variant called BriS Δ affects ACE2 binding and infectivity. In addition, they identified Linoleic acid (LA) as the small molecule that tightly binds to the previously identified, free fatty acid (FFA) binding pocket in the locked structure of the SARS-cov-2. The furin recognition site and S1/S2 cleavage site are located in a flexible loop which is shorter and more rigid in BriS Δ due to deletions. Their Cryo-EM structure of BriS Δ showed particles in locked conformation where RBD is in the closed conformation. They used targeted MD (TMD) to characterize the conformational changes in WT and BriS Δ to discover the force needed to open a single RBD for both apo and LA-bound systems. They found that LA-bound substantially reduces the number of opening events in the WT compared to apo WT spike. Interestingly, LA-bound BriS Δ showed no opening with this force constant and they had to increase the force constant for observing the opening in this variant. They concluded that LA binding stabilizes the locked conformation of both WT and BriS Δ variant. They further studied the effect of LA removal on conformational changes and allostery in the spike protein using dynamical-nonequilibrium simulations. Through this they showed allosteric communication between the LA pocket and V622-L629, D808-S813 and the FPPR (residues 833-855) in WT. In BriS Δ however, only changes in conformation of V622-L629 was observed and only small rearrangements were observed in the deletion site confirming the more rigid structure of shortened loop. I have the following major and minor comments:

Major comments:

1. In a TMD, a time-dependent holonomic constraint is used on the RMSD to a target structure and perturbations imposed by TMD are linear in time. on the other hand, in steered MD, a harmonic potential restrains the variable around the reference value which is moved to the target with a constant rate. So by definition, in TMD due to use of a holonomic constraint reaching the target configuration is guaranteed (In some of the TMD results reported here the target structure is not reached e.g. fig 4.g). However, in some recent works TMD is also associated with RMSD process with harmonic restraint rather than holonomic constraint. Can you please specify if a constraint or a restraint was used in the TMD simulations? Also as far as I know TMD is not implemented in Gromacs. Can you please comment on whether additional software were used for the TMD simulations?
2. In TMD, a cross-over was defined for the target open state of RBD, rather than a percentage of RMSD to the open conformation. This could lead to situations where a cross-over happens but the RBD could still be in an intermediate conformation and not the open state after 10ns of simulation which could change the percentage of opening and closing events. Can this definition of cross-over affect the results?

Minor comments:

1. Can you specify how the RMSD was calculated in the TMD simulations? Does it correspond to a single RBD to all the spike protein?
2. The simulated spike protein was non-glycosylated. Does the authors have comments on how the glycosylation pattern could affect the results of TMD calculations? Does the force required to open the RBD in LA bound locked state change upon glycosylation of nearby regions?
3. How was the force constant chosen for TMD simulations? Also is 10ns enough in all cases to stabilize the RMSD values?

4. In a targeted MD the perturbation causes MD to not reach an equilibrium and violates the microscopic reversibility at fast timescales. Due to this lack of reversibility some properties such as the sequence of events cannot be accurately described in this approach. Does this affect the distribution of crossover events in binning of TMD simulations?

5. Berendsen barostat was used for pressure-coupling in the production run. However it was shown that application of this barostat in production step can cause unrealistic temperature gradients [1]. Did the authors observe any large scale temperature fluctuations in the production run?

6. Two motifs in LA-bound locked S trimer appear in BriSΔ. A H-bond cluster by R1031 and the cation- π interactions between R634 and Y829. Does the nonequilibrium MD show breaking of H-bond and π stacking interactions upon removal of LA? How does this removal affect the unfolding of the regions of Y829?

7. How does the allostery occurs between the LA binding pocket and a distant site on furin cleavage domain? Are there any changes in the volume of the binding pocket?

[1] Feller, Scott E., et al. "Constant pressure molecular dynamics simulation: the Langevin piston method." *The Journal of chemical physics* 103.11 (1995): 4613-4621.

Reviewer #2:

Remarks to the Author:

The manuscript from Gupta et al. contains a series of experiments which examine a SARS-CoV-2 virus variant which was generated during laboratory culture, in which the furin cleavage site and surrounding areas of spike are deleted. The experiments presented involve the separation of this virus containing the deletion from a mixed population, followed several virological cell culture infection studies to attempt to understand the replicative consequences of this variant. This is accompanied by the determination of a structure of this variant spike by cryoEM and biophysical characterisation of its binding behaviour. This is then followed by molecular dynamics simulations comparing the wild-type and variant spikes.

There are some interesting conclusions which can be obtained from the results in this manuscript, although I have a few reservations about the validity of several of the virological assays and consequently the conclusions drawn from them. The cryoEM studies are carried out in a highly competent fashion, however, many of the observations made about the structure obtained are not novel and were seen in several earlier published studies, including from the authors themselves. I am not expert in Molecular Dynamics and can therefore not comment on the validity of these experiments. I would suggest an overall shortening of the paper, in line with the number of conclusions that can be drawn. I detail a number of points below:

Virological Assays

- The assays presented in figure 1 are all based on an incorrect method used to quantify the input virus. Viral stocks are quantified by RT-qPCR, which is a measure of number of genome copies and not infectious virus. This measure does not account for possible differences in particle/genome number: infectivity ratios between virus variants. This is even highlighted by the authors (Lines 310-311) where they hypothesise that the deletion variant may produce a more stable virus particle. This method of virus quantification precludes the direct comparison of the two variants. The correct method would be to quantify based on an infectivity assay such as a plaque assay or TCID50. The experiments could also be expressed in terms of multiplicity of infection (MOI) which is the correct form for this sort of assay. A time-course replication study could also be much more informative than a simple dilution.

- The infectivity assays in Fig. 1 also show some interesting characteristics of the cells which are not discussed. Some cells never reach full infection even with undiluted virus. This suggests a difference in permissivity of infection of these cell lines. Some assays also could benefit from being in a better experimental dynamic range.
- The neutralisation assays and their conclusions appear sound, as these experiments compare to a control and are generally less sensitive to particle: infectivity ratios.

cryoEM structure:

- As mentioned above I think the structure in the manuscript is well obtained, however, there is a lot of discussion about structural features which were present in the authors previous structure (PDB: 6zb5) but which were not commented on at the time. This is particularly the section of lines 149-177 which is a discourse on several general structural features observed both in this and previous studies, which should be removed, as none of these add to the conclusions of the paper. The structural interpretation should be shortened. The main, and only, conclusion of this structure, in my opinion, is that they are identical apart from the lack of an open form of the structure in this deletion variant, which suggests that changes in the region of the deletion can affect the spike opening, which is some distance away.

General points:

- In the deletion variant is there a possibility for an S1/S2 cleavage i.e. a lysine or arginine in the disordered loop?
- Please revert your residue numbering throughout the paper to the original virus strain without deletion. This is the convention for highly changeable viruses, otherwise comparison between strains and structures becomes highly frustrating.
- Lines 44-45 and 49-51: there is a bit too much emphasis put on the role of the S2' site and S1 shedding, which are both poorly understood topics. They have, however, been repeated many times in the literature and people therefore assume they are established facts.
- Line 53: the conclusions of this paper were more opening of the spike on cleavage and therefore a possible increase in binding avidity.
- Lines 65-69: There is a lot of overinterpretation here and in other places linking differences in continuous cell lines to organismal niches. This detracts a bit from the potentially interesting mechanistic points.
- Lines 128-129: Please remove discussion of RGD motifs, as the findings of the cited paper for virus infection are speculative. The discussion of these motifs add nothing to the manuscript.
- Line 134: This fatty acid is also present in the density of many other structures. Please reference them.
- Lines 139-142: This is highly repetitive of the authors previous study.
- Line 202: Referring to this as a surrogate neutralisation kit is a bit confusing, as it is not used as a surrogate for a neutralisation. ACE2 binding ELISA would be more appropriate.
- Lines 207-208: Is this difference significant from the previous measurement of 1.4nM, as the error is quoted as $2.5 \text{ nM} \pm 1$. If not then please change to not significantly different.
- Line 216: Please refer to another paper which found a fully closed structure able to bind ACE2 with little change in affinity (Wrobel et al., 2021: <https://doi.org/10.1038/s41467-021-21006-9>)
- Supp table 1: Exposure should be the full dose not a single frame.

Reviewer #3:

Remarks to the Author:

The SARS-CoV-2 variants with higher transmissibility and pathogenicity are complicating current situation of COVID-19 pandemic. Uncovering the molecular mechanisms underlying these altered viral properties is critically important as for designing countermeasures against these SARS-CoV-2 variants. This manuscript presents the alter infectivity and its structural basis for BriSΔ, a SARS-CoV-2 variant with furin cleavage site deletion.

In this manuscript, the authors investigated the infection efficiency of BriSΔ, which was compared and other variants with furin cleavage site mutations and the WT virus. The authors further solved the cryo-EM structure of BriSΔ, and performed MD simulations to explore the links between the furin site deletion and the dynamic characteristics and S protein functions. The data showed that BriSΔ acquired better replication capacity in Vero E6 cells, but its replication was impaired in Calu-3 cells. The MD simulations revealed the different allosteric and dynamical behavior of BriSΔ from the WT S.

In general, these are quite interesting information, but there are a few questions that are needed to be elucidated:

1. The manuscript tends to imply that the BriSΔ was generated as a result of "intra-host" evolution, and that the deletion of furin cleavage site is an adaptation to the deficiency of TMPRSS2. Evidence of low expression level of TMPRSS2 in the respiratory cells in the patient from whom the clinical sample containing BriSΔ, or diverse expression levels of TMPRSS2 in different types of respiratory cells, or in lungs of different individuals would strengthen this concept.
2. What are the infectivity and transmissibility of BriSΔ in clinical settings or animal models compared to WT SARS-CoV-2 and other variants of concern?
3. How is the density of the 676-682 loop in the C3 structure? In addition, the structural difference from the WT caused by deletion of the 8 residues in BriSΔ should be described more clearly. For example, the H-bond cluster formed by R1031 the cation-π interactions between R634 and Y829 in the structure of BriSΔ were described, but what are like in the WT S structure?
4. In supplementary Fig. 9, the time axis for the long grey rectangle shows the time scale for the unperturbed equilibrium simulation was 200 ns, but the legend for the grey rectangle indicates the simulation was 300 ns, so was the text in line 112, so how long on earth did the unperturbed simulations were performed? The start conformations for the nonequilibrium perturbed simulations were extracted after 50 ns of the unperturbed simulations. Does it mean that the unperturbed simulations reached equilibrium after 50 ns? RMSD plots for the protein in the whole unperturbed simulations should be provided to show when the simulations reached equilibrium.
5. Cryo-EM and MD simulations suggested that the closed conformation is stabilized as compared to WT S by the additional interactions we described within the glycoprotein trimer. The relationship between the stabilization of the closed conformation and the virus infectivity can be discussed.

Reviewer #1 (Remarks to the Author):

In this manuscript the authors used a variety of experimental and MD simulation techniques to understand how deletion of 8 amino acid residues in the furin recognition motif and S1/S2 cleavage site of SARS-Cov-2, a variant called BriS Δ affects ACE2 binding and infectivity. In addition, they identified Linoleic acid (LA) as the small molecule that tightly binds to the previously identified, free fatty acid (FFA) binding pocket in the locked structure of the SARS-cov-2. The furin recognition site and S1/S2 cleavage site are located in a flexible loop which is shorter and more rigid in BriS Δ due to deletions. Their Cryo-EM structure of BriS Δ showed particles in locked conformation where RBD is in the closed conformation. They used targeted MD (TMD) to characterize the conformational changes in WT and BriS Δ to discover the force needed to open a single RBD for both apo and LA-bound systems. They found that LA-bound substantially reduces the number of opening events in the WT compared to apo WT spike. Interestingly, LA-bound BriS Δ showed no opening with this force constant and they had to increase the force constant for observing the opening in this variant. They concluded that LA binding stabilizes the locked conformation of both WT and BriS Δ variant. They further studied the effect of LA removal on conformational changes and allostery in the spike protein using dynamical-nonequilibrium simulations. Through this they showed allosteric communication between the LA pocket and V622-L629, D808-S813 and the FPPR (residues 833-855) in WT. In BriS Δ however, only changes in conformation of V622-L629 was observed and only small rearrangements were observed in the deletion site confirming the more rigid structure of shortened loop.

We thank the Reviewer for this positive assessment of our work.

I have the following major and minor comments:

Major comments:

1. In a TMD, a time-dependent holonomic constraint is used on the RMSD to a target structure and perturbations imposed by TMD are linear in time. on the other hand, in steered MD, a harmonic potential restrains the variable around the reference value which is moved to the target with a constant rate. So by definition, in TMD due to use of a holonomic constraint reaching the target configuration is guaranteed (In some of the TMD results reported here the target structure is not reached e.g. fig 4.g). However, in some recent works TMD is also associated with RMSD process with harmonic restraint rather than holonomic constraint. Can you please specify if a constraint or a restraint was used in the TMD simulations? Also as far as I know TMD is not implemented in Gromacs. Can you please comment on whether additional software were used for the TMD simulations?

Our response: For the TMD simulations, a harmonic restraint, and not a holonomic constraint, was applied and we have clarified this in the manuscript. As such, the TMD does not guarantee that the target conformation will be reached. To answer the second question: we did not need to use any software

other than GROMACS for these position-restrained TMD simulations. This has been clarified in the revised version of our manuscript (see below).

2. In TMD, a cross-over was defined for the target open state of RBD, rather than a percentage of RMSD to the open conformation. This could lead to situations where a cross-over happens but the RBD could still be in an intermediate conformation and not the open state after 10ns of simulation which could change the percentage of opening and closing events. Can this definition of cross-over affect the results?

Our response: We thank the Reviewer for this interesting question. Firstly, it is worth noting that the ACE2 receptor can access the RBD in a less extended conformation than in the open Cryo-EM structure. We are citing the paper by Mert Gur *et al.*, Conformational transition of SARS-CoV-2 spike glycoprotein between its closed and open states, *J. Chem. Phys.* **153**, 075101 (2020), in the revised version of our manuscript. Any opening events in which the RBD was closer to the open than the closed state (i.e. greater than 50% open) was counted as open, because it could still feasibly accommodate the ACE2 interaction. Secondly, our aim here is to compare the behaviour of the WT and Bris Δ spike proteins. Therefore, we used exactly the same restraints and RMSD calculations to compare the WT and Bris Δ (apo and LA-bound) spikes and compare their opening behaviour. The relative differences reflect the relative stability of the different forms, i.e. afforded by either the mutated furin site or the bound LA, or both. This has been clarified in the revised version of the manuscript (see below).

Minor comments:

1. Can you specify how the RMSD was calculated in the TMD simulations? Does it correspond to a single RBD to all the spike protein?

Our response: Yes, the RMSDs were calculated for the C-alpha positions for the respective single RBD for each case. We have added further description to the TMD section in the revised version of our manuscript to address major points 1,2 and minor point 1:

RMSDs were calculated for C-alpha positions corresponding to the single RBD in each case. Openings in which this RBD moved closer to the open than closed state (i.e. greater than 50% open) were counted as open, as this conformation could still feasibly accommodate an ACE2 receptor interaction. As the same harmonic restraints and RMSD calculation methods were used to compare the WT and Bris Δ systems (apo and LA-bound), the relative differences reflect the degree of stability afforded by either the mutated furin site, the bound LA, or both.

2. The simulated spike protein was non-glycosylated. Does the authors have comments on how the glycosylation pattern could affect the results of TMD calculations? Does the force required to open the RBD in LA bound locked state change upon glycosylation of nearby regions?

Our response: Here, we simulate the structures solved by Cryo-EM. As these are expressed in insect cells, they lack glycosylation. The aim of the simulations is to examine the behaviour of precisely these experimentally determined structures. The effects of glycosylation on the spike in vivo are clearly of interest. We have examined effects of full glycosylation on some properties of the WT spike in previous work (*Biophysical Journal* 120, 983 (2021)). The referee raises an interesting question, but the definitive answer to this question is beyond the scope of this paper. Glycosylation may affect the force required for opening. However, when comparing the WT with Bris Δ there are no changes in glycosylation sites,

so the effects of glycosylation will be the same for both and will not significantly alter the conclusions here.

We have added the following text to the end of the TMD section in the revised version of our manuscript:

The spike protein is glycosylated and though glycosylation probably influences the ease with which the RBD opens, the glycosylation sites of BrisΔ are the same as WT and so the effects exerted by glycosylation are likely to be similar for both cases.

3. How was the force constant chosen for TMD simulations? Also is 10ns enough in all cases to stabilize the RMSD values?

Our response: Again the referee raises a good and important question, and we have added detail to the revised version of our manuscript to describe this in the TMD section of our revised manuscript:

Initially a range of harmonic force restraints was applied to the WT spike system to find a restraint force under which the RBD opened in half of the test 10 ns simulations. Such conditions allowed an appropriately large number of repeats to be statistically significant over these nonequilibrium simulations. Thus, we performed 50 simulations over 10 ns for each system (Fig. 4a-d) applying a harmonic restraint force constant of 0.2 kJ/mol/nm to raise a single RBD from an equilibrated, closed conformation to a target, open state, corresponding to the energy-minimized open S model built on EMD-1146²³.”

4. In a targeted MD the perturbation causes MD to not reach an equilibrium and violates the microscopic reversibility at fast timescales. Due to this lack of reversibility some properties such as the sequence of events cannot be accurately described in this approach. Does this affect the distribution of crossover events in binning of TMD simulations?

Our response: As the referee points out, these are nonequilibrium simulations. As noted above, our aim here is to compare the WT and BrisΔ spikes under the same conditions, to identify differences in behaviour that may be of functional relevance. We do not aim to calculate free energy barriers because these are nonequilibrium simulations. The frequency of openings under these conditions is, however, likely to reflect the relative free energy barrier to opening. The results show a meaningful and significant difference in opening propensity between the WT and BrisΔ spikes.

5. Berendsen barostat was used for pressure-coupling in the production run. However, it was shown that application of this barostat in production step can cause unrealistic temperature gradients¹. Did the authors observe any large-scale temperature fluctuations in the production run?

Our response: The Berendsen barostat was chosen for its stability (also found in our previous simulations; we follow a consistent approach here) and we did not see any large-scale temperature fluctuations. (Temperature fluctuations were less than 1% over the 10 ns runs).

6. Two motifs in LA-bound locked S trimer appear in BriSΔ. A H-bond cluster by R1031 and the cation-pi interactions between R634 and Y829. Does the nonequilibrium MD show

¹ Feller, Scott E., et al. "Constant pressure molecular dynamics simulation: the Langevin piston method." The Journal of chemical physics 103.11 (1995): 4613-4621.

breaking of H-bond and pi stacking interactions upon removal of LA? How does this removal affect the unfolding of the regions of Y829?

Our response: No significant difference was observed for the cation- π interaction between R634 and Y829 in BriS Δ between the equilibrium and nonequilibrium MD simulations (Supplementary Fig. 16). Additionally, the number of unstructured residues in the region surrounding Y829 (D800-E860) in BriS Δ was determined for the equilibrium and nonequilibrium MD simulations. Overall, no increase in the number of unstructured residues was observed during the 5 ns following LA removal. In the revised version of our manuscript, we have added a new Figure to the Supplementary Information (Supplementary Fig. 16) to show this, and have added a sentence to the manuscript, noting that no breaking of the R634-Y829 cation- π interaction or unfolding of the Y829 region was observed in BriS Δ upon LA removal from the FFA sites:

Additionally, no breaking of the cation- π interaction between R634 and Y829 (Supplementary Fig. 16) or unfolding of the Y829 region was observed in BriS Δ upon LA removal.

On the question about the network of interactions formed by R1031 in the BriS Δ : no significant difference was observed in the number of hydrogens bonds formed by R1031 in the equilibrium and nonequilibrium MD simulations (Supplementary Fig. 17). Moreover, no breaking of the R1031-F1034 cation- π interaction and R1031-E1023 salt-bridge was observed in the nonequilibrium simulations (Supplementary Fig. 18). Two new figures, namely Supplementary Figs. 17 and 18, have been added to the Supplementary Information in the revised version of our manuscript, and a sentence added to the manuscript explaining that no major change was observed in the network of interactions of R1031 upon LA removal:

Similarly, no significant changes were observed in the network of interactions formed by R1031 in the 5 ns following LA removal (Supplementary Figs. 17 and 18).

7. How does the allostery occurs between the LA binding pocket and a distant site on furin cleavage domain? Are there any changes in the volume of the binding pocket?

Our response: Due to the resolution (i.e. rate of frame saving) of the equilibrium and nonequilibrium trajectories (100 ps), it is not possible to plot the very rapid initial steps (occurring in the first tens of picoseconds) in detail. After 100 ps (the first conformation written in the trajectory file after LA removal), the signal has already reached the furin-cleavage region (Fig. 5 and Figure R1). Storage limitations preclude saving coordinates more frequently. Nevertheless, we can infer pathways for signal propagation from the FFA pocket to the furin-cleavage region from the data. Given that the C525-K537, F318-I326, and L629-Q644 segments also show a structural response to LA removal, signal transmission from the FFA site to the furin site is likely to occur via these regions (Figure R1).

Figure R1- Apparent pathway connecting the FFA pocket to the furin-cleavage region in the WT. A similar communication pathway is predicted for the BriSΔ spike.

Regarding the FFA binding pocket: in both the WT and BriSΔ spikes, it contracts upon LA removal (see Supplementary Figs. 13, 14 and 15), collapsing and becoming occupied by the sidechains of the residues that form the pocket. Supplementary Figs. 13 and 14 show that the distance between the centre of mass of E406-I358, F377-Y369 and I434-L368 is reduced in the nonequilibrium simulations compared to the equilibrium ones. We have added a new sentence to the revised version of the manuscript stating that the FFA binding pocket contracts upon LA removal, and referring to these additional illustrative figures):

Upon LA removal, the FFA pocket contracts becoming occupied by the sidechain of the residues lining the pocket (Supplementary Figs. 13, 14 and 15).

We have also added three new figures to the Supporting Information (Supplementary Figs. 13, 14 and 15), two showing the histograms of the distances between the residues lining the pockets (Supplementary Figs. 13 and 14) and one with an example of the FFA pocket in the beginning and end of a nonequilibrium simulation of BriSΔ (Supplementary Fig. 15).

We thank this reviewer for the helpful comments which allowed us to improve our manuscript.

Reviewer #2 (Remarks to the Author):

The manuscript from Gupta et al. contains a series of experiments which examine a SARS-CoV-2 virus variant which was generated during laboratory culture, in which the furin cleavage site and surrounding areas of spike are deleted. The experiments presented involve the separation of this virus containing the deletion from a mixed population, followed several virological cell culture infection studies to attempt to understand the replicative consequences of this variant. This is accompanied by the determination of a structure of this variant spike by cryoEM and biophysical characterisation of its binding behaviour. This is then followed by molecular dynamics simulations comparing the wild-type and variant spikes.

There are some interesting conclusions which can be obtained from the results in this manuscript, although I have a few reservations about the validity of several of the virological assays and consequently the conclusions drawn from them. The cryoEM studies are carried out in a highly competent fashion, however, many of the observations made about the structure obtained are not novel and were seen in several earlier published studies, including from the authors themselves. I am not expert in Molecular Dynamics and can therefore not comment on the validity of these experiments. I would suggest an overall shortening of the paper, in line with the number of conclusions that can be drawn. I detail a number of points below.

Virological Assays

- The assays presented in figure 1 are all based on an incorrect method used to quantify the input virus. Viral stocks are quantified by RT-qPCR, which is a measure of number of genome copies and not infectious virus. This measure does not account for possible differences in particle/genome number: infectivity ratios between virus variants. This is even highlighted by the authors (Lines 310-311) where they hypothesise that the deletion variant may produce a more stable virus particle. This method of virus quantification precludes the direct comparison

of the two variants. The correct method would be to quantify based on an infectivity assay such as a plaque assay or TCID50. The experiments could also be expressed in terms of multiplicity of infection (MOI) which is the correct form for this sort of assay. A time-course replication study could also be much more informative than a simple dilution.

Our response: We agree with the Reviewer that virus infection and growth analysis experiments should be done using a defined MOI based on the infectivity of a virus on a specific cell type and we routinely use this method. However, this assumes that the infectivity of different virus particles equates with equal numbers of infectious particles and the same efficiency of infection on different cell types, which we believe is not the case with the WT and Bri Δ viruses. Indeed, we have previously shown that the WT and Bri Δ viruses differ in their infection profiles on different cell types (see Refs Daly J.L. et al Neuropilin-1 is a host factor for SARS-CoV-2 infection. *Science* **370**, 861-865 (2020); Peacock, T.P. et al. The furin cleavage site in the SARS-CoV-2 spike protein is required for transmission in ferrets. *Nat Microbiol* (2021), both cited in the revised version of our manuscript) and fitness in Vero E6 and Caco-2/ACE2 cells as shown in Figure 1a. The difference in infectivity on different cell types is likely influenced by differences in the structures of the cleaved and non-cleaved Spikes, the route of entry (ie fusion at the plasma membrane or entry via receptor mediated endocytosis), cell cofactors (ie neuropilins) and interferon stimulated gene products that may inhibit virus infection depending on the route of entry and cell type. Therefore, comparing the infectivity of the two viruses on multiple cell types using infection conditions/MOI based on infectivity on a specific cell type does not necessarily equate to infection with the same number of infectious particles, as is typically the case. In the original version of the manuscript, we tried to balance the amounts of the two viruses used for infection on viral genome copy numbers rather than using a MOI based on the infectivity on a specific cell type (as has been done in some studies), which clearly shows that the two viruses differ in infectivity on the different cell types.

However, we agree with the Reviewer that the genome copy numbers may also not reflect infectious particles or particle numbers. We have therefore produced new data for the revised version of our manuscript to support the findings shown in Figure 1 (now included as Supplementary Fig 2). As requested by the Reviewer we have conducted a time course replication study for the two viruses on Vero E6 and Caco-2/ACE2 cells, which clearly showed differences in their ability to be infected by the two viruses (as shown in Figure 1b and e). For this experiment we first determined the infectivity of the two viruses on the two cell lines and based on these titres (which are clearly different) then did reciprocal infections on the two cell lines using a MOI of 0.5 to allow virus cell-to-cell spread. VeroE6 cells were infected based on the titres of the viruses determined on either Vero E6 cells or Caco2/ACE2 cells and vice versa. We then quantitated the infection on the cells by measuring the amount of the viruses released from the cells and intracellular staining for the viral N protein over the period 24 – 72 hours post-infection. The results clearly show that the Bri Δ virus infects VeroE6 cells more efficiently than the WT virus when the MOI is based on either infectivity from VeroE6 cells or Caco2/ACE2 cells, whilst the infection on the Caco2/ACE2 cells is significantly different when the VeroE6 cell titre is used to calculate the MOI but not the Caco2/ACE2 cell titre. It should be noted however that to balance the Caco2/ACE2 MOIs, different amounts of the viruses have to be used (based on genome copy numbers) which reflects their inherent differences in infectivity on this cell type, when comparing infectious particle to infectious particle. The new supplementary data supports our previous findings and the conclusion that the Bri Δ mutation alters infectivity in a cell-type specific fashion.

To complement the data in Supplementary Fig.2 and explain why we used amounts of virus based on genome copy number we have added a section in the text and modified the text in the original version as follows:

The differences in viral fitness on the two cell lines was supported by viral growth assays which showed that the infectivity of the two viruses on Vero E6 cells did not equate to the

same infectivity on Caco-2-ACE2 cells when infecting with a multiplicity of infection (MOI) based on a Vero E6 cell infection titre and vice versa (Supplementary Fig. 2). As such, differences in the infectivity of equal amounts of the WT and BrisΔ viruses, estimated on genome equivalence by qRT-PCR (equating to a starting MOI of 10 for the WT virus, based on the Vero E6 cell titer) was then compared on Vero E6, Vero E6/TMPRSS2, Caco-2, Caco-2-ACE2 and Calu-3 cells using a range of virus dilutions for infection (Fig. 1b-f).

- The infectivity assays in Fig. 1 also show some interesting characteristics of the cells which are not discussed. Some cells never reach full infection even with undiluted virus. This suggests a difference in permissivity of infection of these cell lines. Some assays also could benefit from being in a better experimental dynamic range.

Our response: We agree with Reviewer #2 that the permissivity of virus infection is an interesting point and not always obvious in studies that express % cell infection relative to maximal infection. We now discuss the difference in permissivity in the revised version of our manuscript as follows:

Differences in the maximal level of infection of the different cell lines at 18 hours after virus infection were observed, most likely due to either differences in the expression levels of ACE2 and cellular proteases required for virus entry or intrinsic cellular factors restricting initial viral replication.

- The neutralisation assays and their conclusions appear sound, as these experiments compare to a control and are generally less sensitive to particle: infectivity ratios.

Our response: We thank and agree with the Reviewer.

cryoEM structure:

- As mentioned above I think the structure in the manuscript is well obtained, however, there is a lot of discussion about structural features which were present in the authors previous structure (PDB: 6zb5) but which were not commented on at the time. This is particularly the section of lines 149-177 which is a discourse on several general structural features observed both in this and previous studies, which should be removed, as none of these add to the conclusions of the paper. The structural interpretation should be shortened. The main, and only, conclusion of this structure, in my opinion, is that they are identical apart from the lack of an open form of the structure in this deletion variant, which suggests that changes in the region of the deletion can affect the spike opening, which is some distance away.

Our response: We agree in principle with the Reviewer that we did not describe these features in our previous, original study, but wish to point out that there was a reason, namely the format and size limitations of the original paper. Although we kept the description brief in our view, we would like to describe these features here as we think this is very valuable information to readers. Moreover, we wish to point out that Reviewer 3 asks us to even further describe differences in structural features in yet more detail, of BrisΔ as compared to WT. We thus believe that a fair compromise is to keep the discussion of these important structural features as they are in the present version of our manuscript, and we thank the Reviewers for their understanding.

General points:

- In the deletion variant is there a possibility for an S1/S2 cleavage i.e. a lysine or arginine in the disordered loop?

Our response: This is a good point raised by the Reviewer. We think there is no possibility for S1/S2 cleavage and there is no lysine and arginine in the disordered loop, and also not in the vicinity of this loop.

- Please revert your residue numbering throughout the paper to the original virus strain without deletion. This is the convention for highly changeable viruses, otherwise comparison between strains and structures becomes highly frustrating.

Our response: As requested by the Reviewer, we have reverted residue numbering corresponding to original virus strain throughout the manuscript, also, the numbering from BrisΔ variant is mentioned in the brackets. As we must have the changed numbering in PDB submission, we have added an explanation regarding this in the data availability section to avoid any confusion for future readers. The numbering in Supplementary Table 2 has been kept unchanged to avoid confusion with PDB submission with an explanation added in the footnotes.

- Lines 44-45 and 49-51: there is a bit too much emphasis put on the role of the S2' site and S1 shedding, which are both poorly understood topics. They have, however, been repeated many times in the literature and people therefore assume they are established facts.

Our response: We agree with the Reviewer. In the revised version of our manuscript, we have altered this sentence as follows:

After intracellular cleavage at the S1/S2 junction by a furin-like protease to produce the S1 and S2 subunits, S gets destabilized and can be further primed by cleavage at the S2' site by host serine proteases on the plasma membrane such as TMPRSS2 or the endosomal cysteine proteases cathepsin B/L.

- Line 53: the conclusions of this paper were more opening of the spike on cleavage and therefore a possible increase in binding avidity.

Our response: We thank the Reviewer for pointing this out. We have altered this sentence in the revised version of our manuscript as follows:

Furin-cleaved S was shown to open more efficiently suggesting an increased binding to human ACE2 than uncleaved S.

- Lines 65-69: There is a lot of overinterpretation here and in other places linking differences in continuous cell lines to organismal niches. This detracts a bit from the potentially interesting mechanistic points.

Our response: We follow the argument of the Reviewer, however, we believe that ours is a valid and important hypothesis supported by data. Nonetheless, we have moved this part to the discussion section in the revised version of our manuscript.

- Lines 128-129: Please remove discussion of RGD motifs, as the findings of the cited paper for virus infection are speculative. The discussion of these motifs add nothing to the manuscript.

Our response: As requested by the Reviewer, we have removed this in the revised version of our manuscript.

- Line 134: This fatty acid is also present in the density of many other structures. Please reference them.

Our response: As requested by the Reviewer, we have added PDBIDs of the other structures with density for fatty acid (modelled or unmodelled) in the revised version of our manuscript.

- Lines 139-142: This is highly repetitive of the authors previous study.

Our response: We thank the Reviewer, however, we wish to point out that here, we further confirmed the LA binding of spike using an orthogonal, unambiguous method we implemented, HILIC-MS-MS. Other manuscripts noting the density, after our original publication, did not provide experimental evidence from an orthogonal method for their interpretation. Our experimental evidence, which provides essential additional proof of the nature of the ligand, strengthens our previous finding of LA binding to spike, and provides confidence to the other reports that have later confirmed our finding, typically based only on the presence of density in the pocket in RBD. We therefore prefer and think it important to present this orthogonal data here.

- Line 202: Referring to this as a surrogate neutralisation kit is a bit confusing, as it is not used as a surrogate for a neutralisation. ACE2 binding ELISA would be more appropriate.

Our response: We respectfully disagree. This is the trademarked name of the kit from manufacturer, and it is an FDA approved test kit. To clarify, we have changed it to ACE2 binding ELISA and added the trademark name in brackets in the revised version of our manuscript.

- Lines 207-208: Is this difference significant from the previous measurement of 1.4nM, as the error is quoted as $2.5 \text{ nM} \pm 1$. If not then please change to not significantly different.

Our response: We agree with the Reviewer. We have changed the text accordingly.

- Line 216: Please refer to another paper which found a fully closed structure able to bind ACE2 with little change in affinity (Wrobel et al., 2021: <https://doi.org/10.1038/s41467-021-21006-9>)

Our response: We have added the reference as per reviewer's suggestion with addition of following sentence:

Also, similar observations were made in a recent study which identified a fully closed Cryo-EM structure of S with no significant change in ACE2 binding affinity.

- Supp table 1: Exposure should be the full dose not a single frame.

Our response: We have changed the exposure to full dose as suggested by reviewer.

We thank the Reviewer for his positive assessment of our work and the very helpful comments and suggestions which we have included in the revised version of the manuscript.

Reviewer #3 (Remarks to the Author):

The SARS-CoV-2 variants with higher transmissibility and pathogenicity are complicating current situation of COVID-19 pandemic. Uncovering the molecular mechanisms underlying these altered viral properties is critically important as for designing countermeasures against these SARS-CoV-2 variants. This manuscript presents the alter infectivity and its structural basis for BriS Δ , a SARS-CoV-2 variant with furin cleavage site deletion.

In this manuscript, the authors investigated the infection efficiency of BriS Δ , which was compared and other variants with furin cleavage site mutations and the WT virus. The authors further solved the cryo-EM structure of BriS Δ , and performed MD simulations to explore the links between the furin site deletion and the dynamic characteristics and S protein functions. The data showed that BriS Δ acquired better replication capacity in Vero E6 cells, but its replication was impaired in Calu-3 cells. The MD simulations revealed the different allosteric and dynamical behavior of BriS Δ from the WT S.

In general, these are quite interesting information, but there are a few questions that are needed to be elucidated:

1. The manuscript tends to imply that the BriS Δ was generated as a result of “intra-host” evolution, and that the deletion of furin cleavage site is an adaption to the deficiency of TMPRSS2. Evidence of low expression level of TMPRSS2 in the respiratory cells in the patient from whom the clinical sample containing BriS Δ , or diverse expression levels of TMPRSS2 in different types of respiratory cells, or in lungs of different individuals would strengthen this concept.

Our response: We concur with the Reviewer that this would be interesting, however it is out of the scope of the current study to analyse TMPRSS2 expression in the patient from which the virus, which was among the first identified by PHE beginning of 2020, was derived.

2. What are the infectivity and transmissibility of BriS Δ in clinical settings or animal models compared to WT SARS-CoV-2 and other variants of concern?

Our response: This is a good point raised by Reviewer. The same viruses used in this study were also used for transmission studies in ferrets. We have already cited and referred to the paper by Peacock, T.P. et al. The furin cleavage site in the SARS-CoV-2 spike protein is required for transmission in ferrets in the revised version of our manuscript.

3. How is the density of the 676-682 loop in the C3 structure? In addition, the structural difference from the WT caused by deletion of the 8 residues in BriS Δ should be described more cleared. For example, the H-bond cluster formed by R1031 the cation- π interactions between R634 and Y829 in the structure of BriS Δ were described, but what are like in the WT S structure?

Our response: We thank reviewer for the pointing this out. The density in the C3 structure is not better than C1, i.e. threefold averaging does not improve the density. The difference in in BriS Δ as compared to WT is that 8 residues from the furin site loop, which is exposed and has a high degree of flexibility in the WT structure, are missing, thus rigidifying the loop in BriS Δ S by restricting flexibility.

4. In supplementary Fig. 9, the time axis for the long grey rectangle shows the time scale for the unperturbed equilibrium simulation was 200 ns, but the legend for the grey rectangle indicates the simulation was 300 ns, so was the text in line 112, so how long on earth did the unperturbed simulations were performed? The start conformations for the nonequilibrium perturbed simulations were extracted after 50 ns of the unperturbed simulations. Does it mean that the unperturbed simulations reached equilibrium after 50 ns? RMSD plots for the protein in the whole unperturbed simulations should be provided to show when the simulations reached equilibrium.

Our response: We thank the reviewer for his/her careful and detailed reading of the manuscript and for spotting this typographical error in the Supplementary Fig. 9 in our original manuscript (please note this is now Supplementary Fig. 10 in our revised manuscript). The equilibrium simulations were indeed performed for 200 ns. Both the Supplementary Figure and its legend have been changed accordingly and are now correct in the revised version of our manuscript.

Furthermore, as requested, the RMSD plots for the unperturbed equilibrium simulations have been added to the supplementary information (Supplementary Fig. 10b). These show that the simulations reached equilibrium after 50 ns. We have also added a sentence to the manuscript stating that all equilibrium simulations were considered sufficiently equilibrated after 50 ns (page 25) based on these data.

5. Cryo-EM and MD simulations suggested that the closed conformation is stabilized as compared to WT S by the additional interactions we described within the glycoprotein trimer. The relationship between the stabilization of the closed conformation and the virus infectivity can be discussed.

Our response: We agree with reviewer, and we expanded our discussion of the relationship between the stabilization of the closed conformation and the infectivity of the virus in the revised version of our manuscript. We write now as follows:

We hypothesize that deletion of the furin cleavage site potentially results in a more stable viral particle secreted from cells, with increased expression of full-length BriSΔ in a prefusion state, which might result in less infectivity of TMPRSS2 expressing cells compared to WT.

We want to thank again all Reviewers for their careful reading and for helpful suggestions which allowed us to improve our manuscript.

Imre Berger, Christiane Schaffitzel, Kapil Gupta

Bristol 24.09.2021

Reviewers' Comments:

Reviewer #1:

Remarks to the Author:

Changes made are sufficient and response to my original criticism strong.

Reviewer #2:

Remarks to the Author:

I am broadly happy with the responses to my comments, however, I have two points which I would like to follow up on further:

Original Question:

The assays presented in figure 1 are all based on an incorrect method used to quantify the input virus. Viral stocks are quantified by RT-qPCR, which is a measure of number of genome copies and not infectious virus. This measure does not account for possible differences in particle/genome number: infectivity ratios between virus variants. This is even highlighted by the authors (Lines 310-311) where they hypothesise that the deletion variant may produce a more stable virus particle. This method of virus quantification precludes the direct comparison of the two variants. The correct method would be to quantify based on an infectivity assay such as a plaque assay or TCID50. The experiments could also be expressed in terms of multiplicity of infection (MOI) which is the correct form for this sort of assay. A time-course replication study could also be much more informative than a simple dilution.

Our response: We agree with the Reviewer that virus infection and growth analysis experiments should be done using a defined MOI based on the infectivity of a virus on a specific cell type and we routinely use this method. However, this assumes that the infectivity of different virus particles equates with equal numbers of infectious particles and the same efficiency of infection on different cell types, which we believe is not the case with the WT and BriSΔ viruses. Indeed, we have previously shown that the WT and BriSΔ viruses differ in their infection profiles on different cell types (see Refs Daly J.L. et al Neuropilin-1 is a host factor for SARS-CoV-2 infection. *Science* 370, 861-865 (2020); Peacock, T.P. et al. The furin cleavage site in the SARS-CoV-2 spike protein is required for transmission in ferrets. *Nat Microbiol* (2021), both cited in the revised version of our manuscript) and fitness in Vero E6 and Caco-2/ACE2 cells as shown in Figure 1a. The difference in infectivity on different cell types is likely influenced by differences in the structures of the cleaved and non-cleaved Spikes, the route of entry (ie fusion at the plasma membrane or entry via receptor mediated endocytosis), cell cofactors (ie neuropilins) and interferon stimulated gene products that may inhibit virus infection depending on the route of entry and cell type. Therefore, comparing the infectivity of the two viruses on multiple cell types using infection conditions/MOI based on infectivity on a specific cell type does not necessarily equate to infection with the same number of infectious particles, as is typically the case. In the original version of the manuscript, we tried to balance the amounts of the two viruses used for infection on viral genome copy numbers rather than using a MOI based on the infectivity on a specific cell type (as has been done in some studies), which clearly shows that the two viruses differ in infectivity on the different cell types.

However, we agree with the Reviewer that the genome copy numbers may also not reflect infectious particles or particle numbers. We have therefore produced new data for the revised version of our manuscript to support the findings shown in Figure 1 (now included as Supplementary Fig 2). As requested by the Reviewer we have conducted a time course replication study for the two viruses on Vero E6 and Caco-2/ACE2 cells, which clearly showed differences in their ability to be infected by the two viruses (as shown in Figure 1b and e). For this experiment we first determined the infectivity of the two viruses on the two cell lines and based on these titres (which are clearly different) then did reciprocal infections on the two cell lines using a MOI of 0.5 to allow virus cell-to-cell spread. VeroE6 cells were infected based on the titres of the viruses determined on either Vero E6 cells or

Caco2/ACE2 cells and vice versa. We then quantitated the infection on the cells by measuring the amount of the viruses released from the cells and intracellular staining for the viral N protein over the period 24 – 72 hours post-infection. The results clearly show that the BriSΔ virus infects VeroE6 cells more efficiently than the WT virus when the MOI is based on either infectivity from VeroE6 cells or Caco2/ACE2 cells, whilst the infection on the Caco2/ACE2 cells is significantly different when the VeroE6 cell titre is used to calculate the MOI but not the Caco2/ACE2 cell titre. It should be noted however that to balance the Caco2/ACE2 MOIs, different amounts of the viruses have to be used (based on genome copy numbers) which reflects their inherent differences in infectivity on this cell type, when comparing infectious particle to infectious particle. The new supplementary data supports our previous findings and the conclusion that the BriSΔ mutation alters infectivity in a cell-type specific fashion.

To complement the data in Supplementary Fig.2 and explain why we used amounts of virus based on genome copy number we have added a section in the text and modified the text in the original version as follows:

The differences in viral fitness on the two cell lines was supported by viral growth assays which showed that the infectivity of the two viruses on Vero E6 cells did not equate to the same infectivity on Caco-2-ACE2 cells when infecting with a multiplicity of infection (MOI) based on a Vero E6 cell infection titre and vice versa (Supplementary Fig. 2). As such, differences in the infectivity of equal amounts of the WT and BriSΔ viruses, estimated on genome equivalence by qRT-PCR (equating to a starting MOI of 10 for the WT virus, based on the Vero E6 cell titer) was then compared on Vero E6, Vero E6/TMPRSS2, Caco-2, Caco-2-ACE2 and Calu-3 cells using a range of virus dilutions for infection (Fig. 1b-f).

Second comment (in response):

I agree that determining an MOI by plaque assay or TCID50 is not an ideal method for these assays, however there is no ideal virological method for this. Determining the MOI is generally the least bad of all the other techniques. You incorrectly assert “this assumes that the infectivity of different virus particles equates with equal numbers of infectious particles and the same efficiency of infection on different cell types”. This is never assumed, by people who understand these assays, for either plaque assays or TCID50 measurements. They are both a measure of a dilution of each virus required for a measurable form of cell infection, nothing to do with particle number. qRT-PCR is a quick and dirty measure of virus genome copy number and is only ever used because it is rapid and facile. This can give you a normal reading for uninfected inactivated virus and can consist of genomes present in your sample from lysed dead cells. You will notice in the two studies that you mention, citing your study, quantify their virus by correct methods. The assumption of a link between infectious particles and qRT-PCR titres is one of the most pervasive fallacies of the COVID-19 pandemic, both in basic and clinical research, and widely pollutes the literature. It is not an acceptable way of measuring virus titre. The inherent problem with the rewrite of the manuscript is that the qRT-PCR titres are viewed as the ground-truth and the infectious titres as an irritating/complicating observation which don't fit with the story being told, whereas the inverse is true.

Supplementary figure 2 added is highly confusing and I am unclear what it is for, as the legend, text and methods do not say what you did, what the experiment was for or what it shows. Fig. S2A: You are not measuring infectious particles/ml, it is infectious titre (which should be in PFU/ml or TCID50/ml). There is also a confusing mixture of mentions of infectious titre, genome equivalence and qPCR in the figure legend and text.

I believe Figure S2A may be a correct experiment (a growth curve) to examine these virus propagation characteristics in different cell types quantified by infection assay. Consequently, it should probably be used as Figure 1 and discussed further.

Original comment:

- As mentioned above I think the structure in the manuscript is well obtained, however, there is a lot of discussion about structural features which were present in the authors previous structure (PDB: 6zb5) but which were not commented on at the time. This is particularly the section of lines 149-177 which is a discourse on several general structural features observed both in this and previous studies, which should be removed, as none of these add to the conclusions of the paper. The structural interpretation should be shortened. The main, and only, conclusion of this structure, in my opinion, is that they are identical apart from the lack of an open form of the structure in this deletion variant, which suggests that changes in the region of the deletion can affect the spike opening, which is some distance away.

Our response: We agree in principle with the Reviewer that we did not describe these features in our previous, original study, but wish to point out that there was a reason, namely the format and size limitations of the original paper. Although we kept the description brief in our view, we would like to describe these features here as we think this is very valuable information to readers. Moreover, we wish to point out that Reviewer 3 asks us to even further describe differences in structural features in yet more detail, of Bris Δ as compared to WT. We thus believe that a fair compromise is to keep the discussion of these important structural features as they are in the present version of our manuscript, and we thank the Reviewers for their understanding.

Second comment (in response):

I am aware about the constrained space for properly describing structures in journals with a single word name and sympathise, however, my main objection was the presentation of these structural features of the Bris(del) spike as novel and therefore related to the deletion in this spike. This is still present in the text: 'We compared our BriS Δ structure with previously determined S structures revealing novel features (Fig. 2)...'. This is not correct and for transparency you need to say that most, if not all, of these features were present in your, and others, previous structures, and are consequently not novel. The lack of discussion of these features in your previous paper, and those of other researchers with near identical structures, do not make them novel. Please reword this section to make it clear what is novel and related to this mutant and truly different from other 'locked' conformation structures, which as far as I can see is only the lack of the 2xPro substitution. Everything else was in previous structures and not related to this deletion. Please state this openly.

We found the Reviewer's insights helpful. Here we provide a point- by-point response to Reviewer #2's comments. You will see that we answered these in full. Changes and additions in the revised version of our manuscript are indicated.

Reviewer #1 (Remarks to the Author):

Changes made are sufficient and response to my original criticism strong.

We thank Reviewer #1 for this very positive assessment of our revised manuscript.

Reviewer #2 (Remarks to the Author):

I am broadly happy with the responses to my comments, however, I have two points which I would like to follow up on further:

We thank Reviewer #2 for this positive assessment of our revised manuscript.

Original Question:

The assays presented in figure 1 are all based on an incorrect method used to quantify the input virus. Viral stocks are quantified by RT-qPCR, which is a measure of number of genome copies and not infectious virus. This measure does not account for possible differences in particle/genome number: infectivity ratios between virus variants. This is even highlighted by the authors (Lines 310-311) where they hypothesise that the deletion variant may produce a more stable virus particle. This method of virus quantification precludes the direct comparison of the two variants. The correct method would be to quantify based on an infectivity assay such as a plaque assay or TCID50. The experiments could also be expressed in terms of multiplicity of infection (MOI) which is the correct form for this sort of assay. A time-course replication study could also be much more informative than a simple dilution.

Our original response: We agree with the Reviewer that virus infection and growth analysis experiments should be done using a defined MOI based on the infectivity of a virus on a specific cell type and we routinely use this method. However, this assumes that the infectivity of different virus particles equates with equal numbers of infectious particles and the same efficiency of infection on different cell types, which we believe is not the case with the WT and Bri Δ viruses. Indeed, we have previously shown that the WT and Bri Δ viruses differ in their infection profiles on different cell types (see Refs Daly J.L. et al Neuropilin-1 is a host factor for SARS-CoV-2 infection. Science 370, 861-865 (2020); Peacock, T.P. et al. The furin cleavage site in the SARS-CoV-2 spike protein is required for transmission in ferrets. Nat Microbiol (2021), both cited in the revised version of our manuscript) and fitness in Vero E6 and Caco- 2/ACE2 cells as shown in Figure 1a. The difference in infectivity on different cell types is likely influenced by differences in the structures of the cleaved and non-cleaved Spikes, the route of entry (ie fusion at the plasma membrane or entry via receptor mediated endocytosis), cell cofactors (ie neuropilins) and interferon stimulated gene products that may inhibit virus infection depending on the route of entry and cell type. Therefore, comparing the infectivity of the two viruses on multiple cell types using infection conditions/MOI based on infectivity on a specific cell type does not necessarily equate to infection with the same number of infectious particles, as is typically the case. In the original version of the manuscript, we tried to balance the amounts of the two viruses used for infection on viral genome copy numbers rather than using a MOI based on the infectivity on a specific cell type (as has been done in some studies), which clearly shows that the two viruses differ in infectivity on the different cell types.

However, we agree with the Reviewer that the genome copy numbers may also not reflect infectious particles or particle numbers. We have therefore produced new data for the revised version of our

manuscript to support the findings shown in Figure 1 (now included as Supplementary Fig 2). As requested by the Reviewer we have conducted a time course replication study for the two viruses on Vero E6 and Caco-2/ACE2 cells, which clearly showed differences in their ability to be infected by the two viruses (as shown in Figure 1b and e). For this experiment we first determined the infectivity of the two viruses on the two cell lines and based on these titres (which are clearly different) then did reciprocal infections on the two cell lines using a MOI of 0.5 to allow virus cell-to-cell spread. VeroE6 cells were infected based on the titres of the viruses determined on either Vero E6 cells or Caco2/ACE2 cells and vice versa. We then quantitated the infection on the cells by measuring the amount of the viruses released from the cells and intracellular staining for the viral N protein over the period 24 – 72 hours post-infection. The results clearly show that the BriSΔ virus infects VeroE6 cells more efficiently than the WT virus when the MOI is based on either infectivity from VeroE6 cells or Caco2/ACE2 cells, whilst the infection on the Caco2/ACE2 cells is significantly different when the VeroE6 cell titre is used to calculate the MOI but not the Caco2/ACE2 cell titre. It should be noted however that to balance the Caco2/ACE2 MOIs, different amounts of the viruses have to be used (based on genome copy numbers) which reflects their inherent differences in infectivity on this cell type, when comparing infectious particle to infectious particle. The new supplementary data supports our previous findings and the conclusion that the BriSΔ mutation alters infectivity in a cell-type specific fashion.

To complement the data in Supplementary Figure 2 and explain why we used amounts of virus based on genome copy number we have added a section and modified the text within our previous version as follows (as indicated in Track Change in the 2nd revision of our manuscript, page 4):

The differences in viral fitness on the two cell lines was supported by viral growth assays which showed that the infectivity of the two viruses on Vero E6 cells did not equate to the same infectivity on Caco-2-ACE2 cells when infecting with a multiplicity of infection (MOI) based on a Vero E6 cell infection titre and vice versa (Supplementary Fig. 2). As such, differences in the infectivity of equal amounts of the WT and BriSΔ viruses, estimated on genome equivalence by qRT-PCR (equating to a starting MOI of 10 for the WT virus, based on the Vero E6 cell titer) was then compared on Vero E6, Vero E6/TMPRSS2, Caco-2, Caco-2-ACE2 and Calu-3 cells using a range of virus dilutions for infection (Fig. 1b-f).

Second comment (in response):

I agree that determining an MOI by plaque assay or TCID50 is not an ideal method for these assays, however there is no ideal virological method for this. Determining the MOI is generally the least bad of all the other techniques. You incorrectly assert “this assumes that the infectivity of different virus particles equates with equal numbers of infectious particles and the same efficiency of infection on different cell types”. This is never assumed, by people who understand these assays, for either plaque assays or TCID50 measurements. They are both a measure of a dilution of each virus required for a measurable form of cell infection, nothing to do with particle number. qRT-PCR is a quick and dirty measure of virus genome copy number and is only ever used because it is rapid and facile. This can give you a normal reading for uninfected inactivated virus and can consist of genomes present in your sample from lysed dead cells. You will notice in the two studies that you mention, citing your study, quantify their virus by correct methods. The assumption of a link between infectious particles and qRT-PCR titres is one of the most pervasive fallacies of the COVID-19 pandemic, both in basic and clinical research, and widely pollutes the literature. It is not an acceptable way of measuring virus titre. The inherent problem with the rewrite of the manuscript is that the qRT-PCR titres are viewed as the ground-truth and the infectious titres as an irritating/complicating observation which don't fit with the story being told, whereas the inverse is true.

Supplementary figure 2 added is highly confusing and I am unclear what it is for, as the legend, text and methods do not say what you did, what the experiment was for or what it shows. Fig. S2A: You are not measuring infectious particles/ml, it is infectious titre (which should be in PFU/ml or TCID50/ml). There is also a confusing mixture of mentions of infectious titre, genome equivalence and qPCR in the figure legend and text. I believe Figure S2A may be a correct experiment (a growth curve) to examine these virus propagation characteristics in different cell types quantified by infection assay. Consequently, it should probably be used as Figure 1 and discussed further.

Our second response: We agree with Reviewer #2 that there is no ideal way to set the infection conditions for the infection assays shown in Figure 1b-f. In our initial revision we provided extra data (Supplementary Figure 2) showing that when the infection conditions are based on MOI values determined using two different cell types, the % of cells infected over time differs (also for different cell types). We firmly agree that the quantitation of viral RNA by qRT-PCR does not necessarily equate with virus infectivity and would not use qRT-PCR as a basis for establishing infection conditions for a comparison of the infectivity of two viruses on a single cell type. In presenting Figure 1 we sought to show that the overall patterns of infectivity for the WT and Bris Δ viruses were markedly different (often in opposing directions) on different cell types, as the starting volumes of the viruses were diluted. We believe this is the case whether the starting volume of virus for the infection assays is based on an MOI or viral genome copies. As such we have retained Fig 1 as sent previously in the first revised version and improved the text and clarity of Supplementary Figure 2 as suggested by Reviewer #2 to avoid ambiguities concerning infection conditions and reinforce the point that there are cell-type specific differences in the infectivity of the WT and Bris Δ viruses which may be due to a number of factors. The changes we made in our 2nd revision are described below.

Main text:

The main text has been altered to explain the decision to use starting virus volumes based on viral genome amounts rather than MOI values as follows:

The differences in the infectivity of the WT and Bris Δ viruses were then compared on Vero E6, Vero E6/TMPRSS2, Caco-2, Caco-2-ACE2 and Calu-3 cells using a range of virus dilutions for infection (Fig. 1b-f). The starting virus volumes for the infections were based on equal viral genome copy numbers as determined by qRT-PCR (equating to a starting multiplicity of infection (MOI) of 10 for the WT virus, based on the Vero E6 cell titer) rather than MOI values. Although viral genome copy numbers do not necessarily reflect virus infectivity, viral growth assays on Vero E6 and Caco-2-ACE2 cells using MOIs determined on either Vero E6 or Caco-2-ACE2 cells, showed that the infectivity of the two viruses differed, depending on the cell type used to determine the MOI (Supplementary Fig. 2).

Supplementary material

As pointed out by Reviewer #2, the Supplementary Fig 2A (now 2B) y-axes were mislabelled. We apologise for this oversight and the confusion it may have caused in interpreting the data. We have now relabelled the y-axes from “infectious particles/ml” to SARS-CoV-2 RNA (genome copies /ml). The y-values have also been changed back to the original genome copies/ml before they were mistakenly calculated as infectious particles/ml. This does not change the shape of the curves or the results.

To improve the flow of Supplementary Fig 2 we have now reordered panel 2B (% of virus infected cells - as detected by staining of the viral N protein) to panel 2A and “the detection of extracellular viral RNA” to panel 2B. The figure legend has been modified as follows:

Supplementary Fig. 2: SARS-CoV-2 WT and Bris Δ growth assays using MOIs determined on different cell types result in differences in infectivity. Caco-2/ACE2 or

*Vero E6 cells were seeded in 96 well plates and infected with two different volumes of the WT and the BriSD virus stocks, each representing a MOI of 0.5 calculated from infectious titres previously determined on either Caco-2/ACE2 (red) or Vero E6 (blue) cells. At the indicated time points post-infection, supernatants were collected from the infected cells, which were then fixed in 4% PFA. (A) The % of virus infected cells was determined by staining the fixed cells using an antibody against the SARS-CoV-2 N protein and DAPI to detect nuclear DNA, before analysis using the ImageXpress Pico automated imaging system. (B) Viral RNA was extracted from pooled culture supernatants from each replicate time point and the amount of viral RNA quantitated by qRT-PCR and expressed as genome copies/ml using a calibrated reference standard. qRT-PCR assays were done in technical triplicate. Statistical significance was determined using a two-tailed unpaired t-test, in which ns = non-significant difference $p < 0.05$, * $p \leq 0.05$, *** $p \leq 0.001$ ($n=6$).*

Supplementary Figure 1 legend. Lines 62-63.

The text was changed to reflect that the point that the viral titre was determined by TCID50 assay (not by qRT-PCR, which was done in addition) and that “infectious” virus particles were serially diluted.

Original comment:

- As mentioned above I think the structure in the manuscript is well obtained, however, there is a lot of discussion about structural features which were present in the authors previous structure (PDB: 6zb5) but which were not commented on at the time. This is particularly the section of lines 149-177 which is a discourse on several general structural features observed both in this and previous studies, which should be removed, as none of these add to the conclusions of the paper. The structural interpretation should be shortened. The main, and only, conclusion of this structure, in my opinion, is that they are identical apart from the lack of an open form of the structure in this deletion variant, which suggests that changes in the region of the deletion can affect the spike opening, which is some distance away.

Our original response: We agree in principle with the Reviewer that we did not describe these features in our previous, original study, but wish to point out that there was a reason, namely the format and size limitations of the original paper. Although we kept the description brief in our view, we would like to describe these features here as we think this is very valuable information to readers. Moreover, we wish to point out that Reviewer 3 asks us to even further describe differences in structural features in yet more detail, of Bris Δ as compared to WT. We thus believe that a fair compromise is to keep the discussion of these important structural features as they are in the present version of our manuscript, and we thank the Reviewers for their understanding.

Second comment (in response):

I am aware about the constrained space for properly describing structures in journals with a single word name and sympathise, however, my main objection was the presentation of these structural features of the Bris(del) spike as novel and therefore related to the deletion in this spike. This is still present in the text: ‘We compared our Bris Δ structure with previously determined S structures revealing novel features (Fig. 2)...’. This is not correct and for transparency you need to say that most, if not all, of these features were present in your, and others, previous structures, and are consequently

not novel. The lack of discussion of these features in your previous paper, and those of other researchers with near identical structures, do not make them novel. Please reword this section to make it clear what is novel and related to this mutant and truly different from other ‘locked’ conformation structures, which as far as I can see is only the lack of the 2xPro substitution. Everything else was in previous structures and not related to this deletion. Please state this openly.

Our second response: We thank Reviewer #2 for pointing out our oversight. We have modified the manuscript main text accordingly and have removed the claim of novelty in our 2nd revision of our manuscript. Specifically, we now write (page 7, main text):

We scrutinized our BriSΔ structure and compared with previously determined S structures for conserved stabilizing features (Fig. 2). Disulfide bonds are known to play a crucial role in stabilizing the S trimer and individual domains.

Moreover, in the following structure description paragraph, we have introduced additional changes (in Track Change in the main text 2nd revision of our manuscript) in response to Reviewer #2’s request.

We thank Reviewer #2 for sharing his/her deep insight and helpful suggestions, which allowed us to introduce further clarity in the 2nd revision of our manuscript.

Additional change:

During our first revision Reviewer #2 requested to change the residue numbering to WT, which we did in the main text but could not yet implement in the PDB submission (as explained in a footnote in previous Supplementary Table 2). In the meantime, with help from the PDB database managers, we could change the residue numbering in our PDB submission to WT; thus we took this opportunity to update the Supplementary Table 2 in this 2nd revision by adjusting the residue numbers and removing the footnote.

Imre Berger, Christiane Schaffitzel, Kapil Gupta

October 24, 2021

Reviewers' Comments:

Reviewer #2:

Remarks to the Author:

I am satisfied with the changes made to manuscript.